Article 

# New molecular components of high and low affinity iron import systems in *Drosophila*

Sattar Soltani [1], Minyi Yan [1], Qingxuan Yu[1], Areeg Abd Elhafiz[1], Erika Pfriem[1], Samuel M. Webb[2], Thomas Kroll[2], Jahir Marceliano Bahena Lopez[3], Fanis Missirlis [3] & Kirst King-Jones [1]✉

The high abundance and molecular versatility of iron have led to its universal presence in biological systems, yet its absorption is exceptionally challenging. Animals and yeasts use divalent metal transporters to import iron, but yeasts also employ the multicopper oxidase Fet3p for high-affinity iron uptake when iron-starved. Using long-term iron depletion in *Drosophila*, we identified four components involved in iron absorption: Multicopper oxidase-4 (Mco4), a Fet3p ortholog, is essential for surviving iron starvation, whereas the cytochrome b561 enzymes Fire (Ferric Iron Reductase) and Fire-like, as well as cytochrome b5 protein Firewood, are required for iron absorption under normal conditions. This study reports the presence of a high-affinity iron uptake system in an animal, a cytochrome b5 electron donor for ferric iron reduction, and intestinal ferric reductases, and provides a valuable resource for further exploration of genes involved in iron homeostasis, transport, and absorption.

Life on Earth depends on iron[1], a redox-active transition metal that donates and accepts electrons in processes such as oxygen transport, detoxification, energy production, and nucleotide synthesis. The vast majority of biological iron is used to produce protein cofactors, i.e., heme and iron-sulfur (Fe-S) clusters[2–4], whereas some proteins bind the metal directly as ionic mononuclear or dinuclear iron[5]. Despite its biological importance, iron has poor bioavailability due to its intrinsic insolubility, particularly in aerobic environments. This low solubility poses a significant barrier to its absorption by organisms. For instance, a typical human adult absorbs only about 2 mg of iron per day, which represents ~0.05% of the body's total iron content.

Not all cells require equivalent amounts of iron. For example, erythroblasts require excess iron to synthesize hemoglobin and differentiate into iron-rich erythrocytes, steroid hormone-producing glands (such as the *Drosophila* prothoracic gland, hereafter "PG"), and tissues involved in detoxification responses (such as liver cells) are also typically iron-rich, because their metabolism relies on high levels of cytochrome P450 enzymes, which require heme cofactors to function.

As such, iron uptake, distribution and sequestration require cell-type-specific regulation to match the iron needs of individual tissues.

Iron pathways are partially conserved between humans and *Drosophila*[6–8]. In vertebrates, duodenal enterocytes are the first site of iron regulation as they govern the uptake and release of absorbed iron[2,8,9]. Duodenal cytochrome B (DCYTB), a transmembrane ferric reductase of the cytochrome b561 (CYB561) family, uses electrons donated by cytoplasmic ascorbate to reduce ferric iron ($Fe^{3+}$) in the gut lumen to ferrous iron ($Fe^{2+}$). DCYTB has two *Drosophila* homologs, CG1275 and Nemy. Of the two, CG1275 appears to be the ortholog of DCYTB, as it has 43% identity compared to 33% identity exhibited by Nemy[10]. DCYTB knockout mice (*Cybrd1*$^{-/-}$) did not show iron deficiency compared to controls, suggesting that either other ferric reductases operate alongside DCYTB[11], or that dietary reductants can compensate for its loss. Besides *CG1275* and *nemy*, six other CYB561 genes are found in *Drosophila* (*CG3592*, *CG8399*, *CG10165*, *CG10337*, *CG13077 = fire-like*, and *CG13078 = fire*)[10]. Of the eight CYB561 proteins, only Nemy has been studied in greater detail, which revealed that *nemy* mutants have

[1]Department of Biological Sciences, University of Alberta, Edmonton, AB, Canada. [2]Stanford Synchrotron Radiation Lightsource, SLAC National Accelerator Laboratory, Menlo Park, CA, USA. [3]Departamento de Fisiología, Biofísica y Neurociencias, Centro de Investigación y de Estudios Avanzados (Cinvestav), Mexico City, Mexico. ✉e-mail: kirst.king-jones@ualberta.ca

reduced memory retention, consistent with the presence of Nemy in synaptic vesicles[12,13]. A recent report described an assay to detect ferric reductase activity in the larval intestinal lumen, but the gene encoding the ferric reductase remains unknown[14].

Once $Fe^{3+}$ is reduced to $Fe^{2+}$ via DCYTB, iron is transported by vertebrate Divalent Metal Transporter 1 (DMT1 aka SLC11A2) into the enterocyte, a process that is likely mirrored in *Drosophila* by the DMT1 ortholog Malvolio[15]. DMT1 is permeable to a wide range of divalent metals and, therefore, cannot efficiently import iron if it is scarce relative to other metals[16]. DMT1 not only plays a role in intestinal iron uptake but is also needed for iron release from late endosomes/lysosomes[17]. Consequently, DMT1 knockout mice (*SLC11A2*[-/-]) die within a week from severe anemia[18]. An intestine-specific DMT1 knockout (*DMT1*[INT/INT]), however, is viable, but animals are anemic, display significantly reduced iron concentrations in various organs, and develop cardiomegaly and other heart problems after six months[19]. These data suggest that DMT1 is critical for iron uptake but does not account for all absorption of non-heme iron in the gut.

Ferrous iron leaves the vertebrate enterocyte via Ferroportin, for which no *Drosophila* ortholog exists. Upon immediate oxidation by Hephaestin or Ceruloplasmin, which represent two of the three vertebrate Multicopper oxidases, ferric iron is loaded onto the serum protein transferrin (Tf) and delivered to target tissues[20]. Iron-loaded Tf binds to the membrane-bound Transferrin Receptor 1 (TfR1) in target tissues, after which the complex is endocytosed, where another ferric reductase allows the release of ferrous iron into the early endosome and its subsequent transfer into the cytosol via a different DMT1 isoform[21–24]. *Drosophila* has three Tf orthologs (Tsf1-3), of which Tsf1 has been shown to deliver iron across the hemolymph to reach target tissues[25]. Curiously, flies lack an ortholog of vertebrate TfR1, suggesting the existence of an uncharacterized insect transferrin receptor.

DMT1 and unrelated proteins with similar functionality in other organisms, such as yeast Fet4p, are sufficient to meet cellular iron requirements under iron-replete conditions despite their low affinity for iron. Under iron-deficient conditions, however, bacteria, fungi, and plants employ two basic strategies to compensate for the inefficiency of low-affinity iron uptake systems. The first strategy uses siderophores, which are secreted iron-chelating molecules that are later recaptured. This process has been observed in bacteria, fungi and graminaceous plants[26,27]. The second strategy utilizes a plasma membrane-bound ferric reductase that reduces extracellular $Fe^{3+}$ to $Fe^{2+}$, and is followed by import into the cell via a high-affinity transport system. In non-graminaceous plants, IRT1, a Zip family member, acts as a high-affinity transporter for $Fe^{2+}$ in roots[28,29]. In the yeast *Saccharomyces cerevisiae*, three proteins ensure high-affinity iron import. The first step is carried out by the ferric reductase Fre1, which converts extracellular ferric iron to ferrous iron. $Fe^{2+}$ is then re-oxidized by Fet3p to $Fe^{3+}$, which is coupled to Ftr1, a permease, which then transports Fet3p-derived $Fe^{3+}$ into the cell. Fet3p is a critical component of this triple-step reduction/oxidation reaction, leveraging distinct iron-binding properties to ensure selective iron uptake[30].

Despite our continuously growing understanding of iron biology and its regulation, our grasp of genome-wide networks that respond to changes in iron levels remains rudimentary. Several microarray studies from various model organisms and cell lines have attempted to characterize the cellular responses to either iron deficiency or iron overload[31–40]. Nonetheless, many of these studies have limitations because they focus solely on the long-term effects of iron overload or deprivation and often examine only a single tissue type. This overlooks the possibility that different tissues might exhibit unique responses to changes in iron levels. Consequently, while end-point measurements provide a straightforward experimental approach, they may fail to capture acute transcriptional responses that occur within hours. We reasoned that one should examine the alimentary canal separately since the gut is the principal site of iron absorption and, therefore, likely has a unique transcriptional profile.

In an earlier study, we showed that mild iron depletion may take multiple generations to elicit a phenotype[41]. In this study, we exploited this treatment to gently deplete wild-type flies of iron over multiple generations. Specifically, we reared flies for five generations on fly media that contained the iron chelator Bathophenanthroline disulfonic acid (BPS) to reduce whole-body iron content to a level that would sensitize animals to sudden increases in dietary iron concentrations, which we accomplished by transferring larvae to fly food supplemented with ferric ammonium citrate (FAC). We then generated RNA-Seq-based gene expression profiles for three different tissue types at four different time points. Lastly, we conducted validation experiments for differentially expressed genes (DEGs) based on genetic and molecular studies of RNAi, mutant and transgenic flies. As source material for the RNA-Seq studies, we used i) the larval brain ring gland complex (BRGC), because it harbors the iron-rich PG, ii) the gut, as it is the site of iron absorption, as well as iii) whole larvae (aka whole body = WB) to monitor genes in all tissues. The data that were derived from these studies revealed genes acting in iron biology, and provide a comprehensive genome-wide resource aimed at expanding our understanding of iron gene networks and their role in iron biology.

## Results

### Trans-generational iron depletion in *Drosophila* larvae

Prior to pupariation, *Drosophila* 3rd instar larvae feed continuously for ~36 h, and require large nutrient inputs to sustain rapid growth[42,43]. When larvae were switched from a normal to an iron-rich diet, we observed only weak to moderate transcriptional changes in known iron-regulated genes (Supplementary Fig. S1A), consistent with the finding that one generation of high iron-feeding did not significantly raise ring gland or CNS iron levels, as assessed by synchrotron-based X-ray fluorescence microscopy (Fig. 1A). Larvae reared on BPS-containing media showed reduced ring gland iron levels but largely unchanged brain iron concentrations (Fig. 1A). These results confirm earlier findings that tissues differ in their iron responses[44], supporting the "sparing model", where the CNS is protected from starvation relative to other tissues[45]. Although iron depletion affected the ring gland within one generation, larval developmental timing was largely unaffected (Fig. 1B), suggesting that iron remained sufficient for gland function.

To minimize potential BPS toxicity or non-specific effects, we used moderate BPS concentrations and maintained fly populations on supplemented media over several generations, gradually lowering systemic iron to critical levels. From generation #2 (G2) onwards, animals pupariated with increasing developmental delays, adding about 4–8 h per generation until G4 (Fig. 1B). Overall survival rates were similar between controls and G1-G3 populations on BPS media. G4 animals showed a moderate drop, and in G5 survival fell sharply (only 42% reached adulthood) with surviving larvae undergoing puparium formation ~32 h later than controls (Fig. 1B). G6 animals fully recovered when transferred to media supplemented with both BPS and FAC (Supplementary Fig. S1B), indicating that iron stores were quickly replenished and that prolonged BPS exposure caused no cumulative toxicity beyond iron chelation.

### RNA sequencing of larval tissues isolated from G6 larvae

After we had established that five generations of iron deprivation caused sufficiently low systemic iron levels to reduce survival rates to <50%, we reasoned that re-feeding with a high-iron diet in G6 would have a more pronounced transcriptional iron response than doing so in earlier generations. We, therefore, continued to rear G6 larvae on BPS media until the L2/L3 molt (~72 h after egg deposition) but then divided animals into two groups, where one was transferred to fresh BPS-supplemented food as a control group, whereas the experimental group was switched to an iron-rich medium (Fig. 1C). To minimize

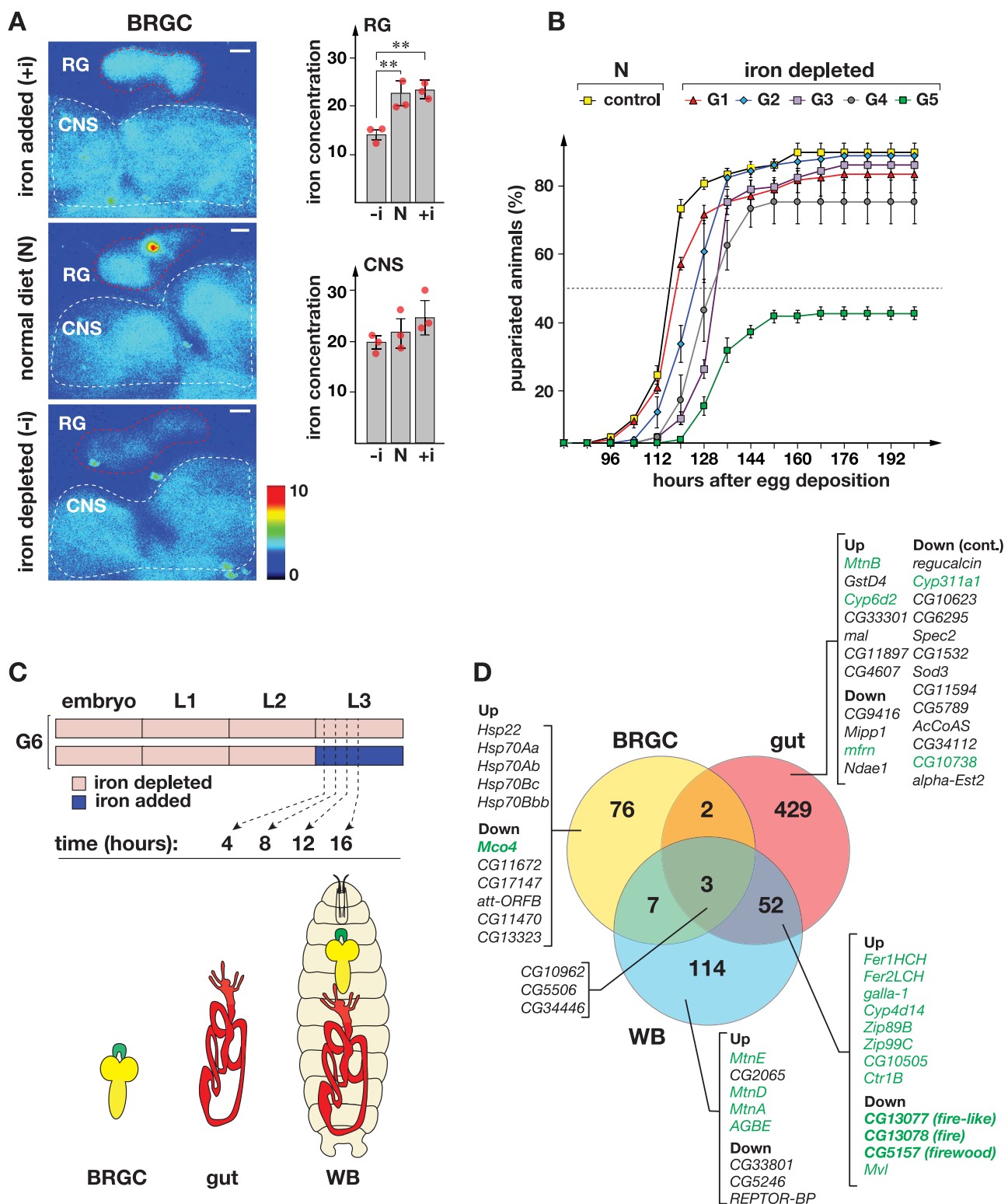

biological variability, we precisely staged all larvae at the L2/L3 molt, i.e., newly formed L3 were transferred within 30 min to either fresh BPS- or FAC-supplemented food. To attain robust gene expression profiles, we analyzed a time course rather than a single time point. To this end, we collected gut, WB and BRGC samples at 4, 8, 12, and 16 h after the L2/L3 molt. Upon RNA sequencing (Illumina Hi-Seq, three lanes), we obtained an average ~22 million paired-end reads per sample. The total number of RNA-Seq samples was 48 (4 time points ×2 media types ×3 tissue types ×2 replicates).

We analyzed the RNA-Seq data using *Arraystar* in conjunction with *MS Access*, which we counterchecked via the R-packages *edgeR* and *DESEq2*[46–48] (Supplementary Data 2). The tables and figures shown in this study are based on the *Arraystar* analysis, unless stated otherwise. Overall, our approach worked very well, evidenced by the high abundance of known iron players (Fig. 1D, green gene names). In total, we identified transcripts of 486 iron-responsive genes in the gut, 176 genes in WB samples, and 88 genes in BRGC samples (Fig. 1D). In brief, the data revealed the expected upregulation of known metal

**Fig. 1 | RNA-Seq strategy to examine genomic response to dietary iron. A** X-Ray Fluorescence Microscopy (XRF) images of BRGC (brain-ring gland complex) samples and quantification of iron levels in the ring gland (RG) and central nervous system (CNS). Larvae were reared for one generation on iron-enriched (+i), normal (N), and in iron-depleted (−i) media, and BRGCs were dissected from L3 larvae. Average iron concentrations in the RG, (red dotted line) and CNS (white dotted line) were calculated from three replicates based on Kα emission/area ratios. Dotted lines indicate regions used to measure iron levels. Scale bars: 30 μm. The color scale bar reflects log2-based iron levels. Asterisks denote significance thresholds ($*p < 0.05$ and $***p < 0.001$). **B** Survival rates and developmental progression of fly populations as a result of multi-generational iron deprivation. $w^{1118}$ flies were reared for five generations (G1–G5) on either a normal diet (N) or media supplemented with BPS. Y-axes denote the percentage of pupariated animals, and X-axes show hours after egg deposition. Dotted line indicates 50% pupariation. Error bars indicate standard deviation from three biological replicates; means are centered. **C** Schematic of the RNA-Seq experimental design. Eggs from G5 flies reared on iron-depleted media (see **B**) were used to produce the G6 generation. G6 larvae were staged within 30 min after the L2/L3 molt and split into two groups: one cohort was transferred to iron-supplemented food (FAC) and the other to fresh iron-depleted media. At 4, 8, 12, and 16 h after the L2/L3 molt, BRGC (Ring gland = green, CNS = yellow), guts (red) and whole larvae (WB = whole body) were collected. L1/L2/L3 denote first, second, and third instar larvae. **D** Venn diagram summarizing RNA-Seq results for a total of 683 differentially expressed genes for i) the BRGC (88 genes), ii) the gut (486 genes) and iii) WB (176 genes). Example genes shown for selected sections. Green: known iron-related genes. Source data are available in the accompanying source data file.

detoxification genes in the gut and whole-body samples and the suppression of genes with hitherto undocumented roles in iron uptake, as well as iron responses in the brain and ring gland. In the next three sections, we provide a detailed breakdown of our findings based on sample type.

## Gut iron response

We expected that the intestine would have the fastest response time to iron since it is the site of nutrient absorption. In total, we identified 486 genes that responded to a switch in dietary iron concentrations. When we further filtered this cohort for genes showing a rapid and sustained response in the gut [a >2-fold change in the first 4 h and showing the same trend (i.e., consistently up or down) in the remaining three time points], we identified 14 upregulated and 38 downregulated protein-encoding genes (Table 1). Of the 14 rapidly upregulated genes, 11 had established links to iron/metals (see Supplementary Data 3 for our iron/metal reference list), and so had 7 of the 38 downregulated genes. Term enrichment statistics for iron/metal genes are depicted in Table 2, which also lists other terms, including enrichment for "cytochrome P450" genes, "ABC transporters" and "transmembrane transport" in the gut data.

From a homeostatic point of view, one would expect to find genes involved iron/metal detoxification to be upregulated when an iron-rich diet is used. By contrast, genes acting in iron uptake and trafficking should be downregulated based on the rationale that such genes must be more highly expressed in an iron-poor environment to compensate for iron scarcity by increasing the capacity for iron capture. Consequently, a switch to a high-iron diet should decrease their expression. Overall, these anticipated trends held true for the gut data. For instance, among the upregulated transcripts were two genes that encode the main ferritin subunits (Fig. 1D, Table 1), ferritin heavy (Fer1HCH) and light (Fer2LCH) chain homologs. Ferritins are molecular nanocages that store excess cytosolic iron, with a capacity of storing hundreds to thousands of oxidized iron atoms per cage[44,49]. Insect ferritins are generally composed of 12 ferritin heavy and 12 light chain proteins[50], consistent with our finding that the genes for both subunits are upregulated on high iron diets.

Another transcript we expected to be rapidly upregulated was Zip99C (also known as Zip13), a member of the ZIP family of transporter proteins[51,52]. Zip13 is tightly linked to ferritin function and is the only identified iron transporter/exporter acting in the ER/Golgi axis. Zip13 is thought to mainly act in the midgut and operates by transferring cytosolic ferrous iron into the ER/Golgi, where iron can be incorporated into ferritin[53]. Consistent with our results, Zip13 showed moderate upregulation in an earlier study (1.5-fold) when animals were fed an iron-rich diet[54]. Due to our prolonged iron deprivation conditions, however, Zip13 displayed a considerably stronger transcriptional response, ranging from 1.9- to 4.4-fold induction (Table 1). Interestingly, we identified a second ZIP family member (Zip89B) among the 14 rapidly upregulated genes. Both Zip89B and Zip99C/Zip13 belong to the SLC39 subfamily, which comprises 14 members in *Drosophila*.

Zip89B has been proposed to act as a low-affinity zinc transporter[55], raising the idea that this transporter may have additional substrates, including iron, and may act in a similar fashion to Zip99C/Zip13, albeit in a different subcellular compartment[56], to facilitate iron detoxification.

Two other upregulated genes, *CG10505* and *MtnB*, which encode an ABC transporter and Metallothionein (Fig. 1D), respectively, are also likely acting in iron detoxification. CG10505 is homologous to several yeast ABC transporters acting in detoxification, including VMR1 (vacuolar multidrug resistance 1) and YCF1 (yeast cadmium factor 1)[57], and has been identified in fly cell culture systems to respond to metals[35]. *Metallothionein B* (*MtnB*), a metal-binding protein previously shown to act in iron detoxification[58]. Additional rapidly upregulated gut transcripts included *galla-1*, which, like its human homolog CIA2A acts in iron-sulfur cluster biosynthesis[59,60], and *CG31288*, which is thought to encode an ecdysteroid kinase. Earlier research indicated that manganese deficiency induces this gene[61].

As explained earlier, we expected that the set of 38 downregulated genes should be enriched for transcripts encoding proteins involved in iron uptake. Consistent with this, we found two well-characterized genes in this cohort, *Malvolio* (aka *Mvl*, the fly ortholog of DMT1) and *mitoferrin* (*mfrn*), which functions in mitochondrial iron uptake[62]. Remarkably, the three most strongly downregulated genes have not been characterized in *Drosophila* and encode two proteins of the cytochrome b561 family (CG13078 and CG13077), as well as a cytochrome b5 protein (CG5157). This was intriguing, since the insect gut ferrireductase(s) responsible for reducing $Fe^{3+}$ to $Fe^{2+}$ were unidentified, but presumed to be either CG1275 or Nemy[8,20]. However, the latter showed no significant response to dietary iron changes, suggesting that CG13078 (hereafter referred to as "Fire") and CG13077 (hereafter referred to as "Fire-like") represent the missing ferrireductases. Given the similarly strong response of CG5157 (hereafter referred to as "Firewood"), we hypothesized that this Cytochrome b5 protein provides the electron required for the reduction of $Fe^{3+}$. We will present functional data on these three genes after we have addressed the genomic response in whole body and BRGC samples.

Our multi-conditional RNA-Seq data permitted us to perform cluster analysis, in order to identify genes that display comparable responses and are thus candidates for being co-regulated. Using this strategy for the 486 genes in the gut cohort, we detected ten clusters (Fig. 2A and Supplementary Data 4). We highlight three clusters that were enriched for previously known iron genes and appeared to be co-regulated in response to our experimental treatments. Cluster #1 comprised eight genes presumably involved in iron uptake, as it included *Malvolio* (the DMT1 ortholog) and *fire, fire-like, firewood* (Supplementary Data 4). This result provides further support to the idea that the function of the *fire* gene complex is linked to dietary iron absorption via Malvolio/DMT1. Cluster #6 (18 genes) represented the immediate response genes to iron treatment, which included genes encoding transporters *Zip48C, Zip89B; Zip89C, CG10505*, the cytosolic iron-sulfur cluster protein *galla-1, MtnB* as well as other highly induced

**Table 1 | Rapid and sustained gene responses in the gut (52 genes)**

| Upregulated (14 genes) | | | | | | |
|---|---|---|---|---|---|---|
| Name | Description | Reported link to metal/iron? | 4 h | 8 h | 12 h | 16 h |
| **MtnB** | Metallothionein | Yes | 13.6 | 3.6 | 10.7 | 3.0 |
| **CG10505** | ABC transporter | Yes (homolog of yeast cadmium factor) | 13.0 | 1.6 | 6.1 | 2.5 |
| CG7763 | Lectin fold / carbohydrate binding | No | 11.2 | 2.1 | 5.2 | 3.5 |
| **galla-1** | Fe-S cluster biosynthesis | Yes | 7.6 | 4.4 | 4.8 | 2.5 |
| Drip | Aquaporin family | Some data links Aquaporin-4 (Drip ortholog) to metal intoxication | 5.0 | 2.3 | 4.9 | 1.8 |
| **Zip99C** | Zinc/iron permease | Yes | 4.4 | 2.5 | 3.6 | 1.9 |
| **Cyp6d2** | Cytochrome P450 6d2 | Yes (harbours heme) | 3.7 | 2.9 | 2.5 | 1.3 |
| **Fer1HCH** | Ferritin 1 heavy chain | Yes (stores iron) | 3.1 | 4.0 | 7.5 | 4.5 |
| GstD2 | Glutathione S transferase | Yes (induced by Zn and Cadmium) | 3.0 | 4.3 | 2.2 | 1.0 |
| CG7720 | Sodium:iodide symporter | No | 2.8 | 2.2 | 3.1 | 1.1 |
| rdog | ABC transporter | Yes (zinc detoxification) | 2.7 | 1.1 | 1.8 | 1.2 |
| **Fer2LCH** | Ferritin 2 light chain | Yes (stores iron) | 2.4 | 3.5 | 8.9 | 4.0 |
| **Zip89B** | Zinc/iron permease | Yes | 2.4 | 1.4 | 3.7 | 3.2 |
| CG31288 | Possible ecdysteroid kinase | Yes (induced by Mn deficiency) | 2.0 | 1.6 | 1.4 | 1.2 |
| Downregulated (38 genes) | | | | | | |
| **Fire (CG13078)** | Ferric reductase | Yes | −45.1 | −7.1 | −40.8 | −78.1 |
| **Fire-like (CG13077)** | Ferric reductase | Yes | −27.3 | −11.3 | −19.3 | −52.1 |
| **Firewood (CG5157)** | Cytochrome b5 | Yes (predicted) | −22.1 | −5.4 | −7.7 | −15.2 |
| CG14205 | acyltransferase | No | −8.0 | −2.0 | −1.8 | −1.4 |
| CG5892 | acyltransferase | No | −7.5 | −2.6 | −1.8 | −3.3 |
| MFS1 | Solute carrier family 17 | No | −5.6 | −2.2 | −1.3 | −1.7 |
| Adh | Alcohol dehydrogenase | No | −5.2 | −2.4 | −1.2 | −1.3 |
| **Mvl** | divalent metal ion transporter | Yes, DMT1 homolog | −4.5 | −2.3 | −2.1 | −3.3 |
| CG9416 | Metallopeptidase | Yes (zinc ion binding) | −4.3 | −2.1 | −1.1 | −2.1 |
| CG11576 | Solute carrier family 52 | No | −4.0 | −1.9 | −1.8 | −2.7 |
| alpha-Est2 | Carboxylesterase | No | −4.0 | −2.8 | −1.3 | −1.3 |
| **mfrn** | Mitochondrial iron import | Yes | −4.0 | −1.6 | −1.1 | −2.4 |
| CG13893 | Phospholipid transport | No | −3.2 | −2.0 | −1.1 | −1.1 |
| Ndae1 | Solute carrier family 4 | No | −3.2 | −1.7 | −1.0 | −1.2 |
| regucalcin | CG1803 | No | −3.1 | −2.5 | −1.3 | −1.1 |
| CG2543 | Folylpolyglutamate synthetase | No | −3.1 | −1.3 | −1.1 | −1.7 |
| **Cyp311a1** | Cytochrome P450 | Yes (heme-binding) | −2.9 | −2.8 | −1.3 | −1.1 |
| CG18179 | peptidase | No | −2.8 | −5.4 | −3.1 | −4.6 |
| CG10623 | Methionine synthesis | Yes (zinc ion binding) | −2.8 | −1.6 | −1.2 | −1.3 |
| CG7589 | Secretory chloride channel | No | −2.7 | −2.1 | −1.3 | −1.6 |
| CG6295 | Triacylglycerol lipase | No | −2.7 | −2.7 | −1.0 | −1.5 |
| Spec2 | small GTPase binding | No | −2.5 | −2.3 | −1.1 | −1.7 |
| Mal-A8 | Solute carrier family 3 | No | −2.5 | −3.2 | −1.2 | −1.2 |
| CG1532 | Glyoxalase | No | −2.5 | −2.2 | −1.5 | −1.8 |
| Sod3 | Superoxide dismutase | Yes (copper/zinc-binding domain) | −2.4 | −2.1 | −1.4 | −1.8 |
| CG11594 | FGGY carbohydrate kinase | No | −2.4 | −1.7 | −1.0 | −1.1 |
| CG5789 | ABC transporter-like | No | −2.3 | −1.4 | −1.2 | −1.8 |
| AcCoAS | acetyl-CoA ligase | No | −2.2 | −1.7 | −1.1 | −2.0 |
| CG34112 | No data | No | −2.2 | −1.8 | −1.1 | −1.0 |
| **CG10738** | CG10738 | Yes (heme NO binding associated) | −2.2 | −2.3 | −1.0 | −1.2 |
| CG8839 | Fatty acid catabolism | No | −2.2 | −1.7 | −1.0 | −1.1 |
| Jon99Fii | endopeptidase | No | −2.1 | −1.9 | −1.5 | −1.3 |
| CG7470 | Glutamate 5-kinase | No | −2.1 | −1.6 | −1.1 | −1.4 |
| CG15773 | No data | No | −2.1 | −2.4 | −1.1 | −1.2 |
| CG17119 | Amino acid transporter | No | −2.0 | −1.6 | −1.1 | −1.2 |
| CG8080 | NAD+ kinase | No | −2.0 | −1.6 | −1.0 | −1.0 |
| CG13654 | No data | No | −2.0 | −2.4 | −1.1 | −1.5 |
| CG33056 | Purine metabolism | No | −2.0 | −1.3 | −1.0 | −1.7 |

The table lists 14 upregulated and 38 downregulated genes in the gut. The gene cohort was obtained by filtering for a >2-fold change at the 4-h time point, combined with a consistent expression trend at the remaining time points. Gene names in bold indicate genes with known or predicted roles in iron metabolism.

**Table 2 | Term enrichment statistics for all gene cohorts**

| Name of cohort (size of cohort out of 14,557 genes) | Metal/iron (839) | | | P450 (89) | | | ABC (57) | | | TM transport (638) | | | Chitin (346) | | | Heat-shock (32) | | |
|---|---|---|---|---|---|---|---|---|---|---|---|---|---|---|---|---|---|---|
| | O | E | P | O | E | P | O | E | P | O | E | P | O | E | P | O | E | P |
| Gut Rapid UP (14) | **8** | **0.8** | **1.5E−16** | 1 | 0.1 | 1.7E−03 | **2** | **0.1** | **8.2E−17** | **6** | **0.6** | **2.0E−12** | 0 | 0.3 | ns | 0 | 0.0 | ns |
| Gut Rapid DOWN (38) | 7 | 2.2 | 8.0E−04 | 1 | 0.2 | ns | 1 | 0.1 | 2.7E−02 | **8** | **1.7** | **5.0E−07** | 0 | 0.9 | ns | 0 | 0.1 | ns |
| Gut UP (77) | 12 | 4.4 | 2.1E−04 | 1 | 0.5 | ns | 2 | 0.3 | 1.9E−03 | 10 | 3.4 | 2.2E−04 | 1 | 1.8 | ns | 1 | 0.2 | 4.3E−02 |
| Gut DOWN (322) | 31 | 18.6 | 2.6E−03 | 5 | 2 | 2.8E−02 | 2 | 1.3 | ns | 23 | 14.1 | 1.4E−02 | 3 | 7.7 | ns | 0 | 0.7 | ns |
| WB Rapid UP (14) | **9** | **0.8** | **5.4E−21** | 0 | 0.1 | ns | 1 | 0.1 | 5.2E−05 | 4 | 0.6 | 9.7E−06 | 1 | 0.3 | ns | 0 | 0.0 | ns |
| WB Rapid DOWN (10) | 4 | 0.6 | 3.4E−06 | 0 | 0.1 | ns | 0 | 0.0 | ns | 1 | 0.4 | ns | 0 | 0.2 | ns | 0 | 0.0 | ns |
| WB UP (53) | **12** | **3.1** | **1.3E−07** | 0 | 0.3 | ns | 1 | 0.2 | ns | 7 | 2.3 | 1.7E−03 | 1 | 1.3 | ns | 0 | 0.1 | ns |
| WB DOWN (41) | 7 | 2.4 | 1.9E−03 | 2 | 0.3 | 4.5E−04 | 0 | 0.2 | ns | 3 | 1.8 | ns | 0 | 1.0 | ns | 0 | 0.1 | ns |
| RG Rapid UP (0) | 0 | 0 | n/a | 0 | 0 | n/a | 0 | 0.0 | n/a | 0 | 0.0 | n/a | 0 | 0.0 | n/a | 0 | 0.0 | n/a |
| RG Rapid DOWN (4) | 0 | 0.2 | ns | 0 | 0 | ns | 0 | 0.0 | ns | 0 | 0.2 | ns | 1 | 0.1 | 3.0E−03 | 0 | 0.0 | ns |
| RG UP (16) | 1 | 0.9 | ns | 0 | 0.1 | ns | 0 | 0.1 | ns | 1 | 0.7 | ns | 1 | 0.4 | ns | **7** | **0.0** | **7.2E−303** |
| RG DOWN (40) | 2 | 2.3 | ns | 0 | 0.2 | ns | 0 | 0.2 | ns | 0 | 1.8 | ns | **14** | **1.0** | **6.6E−42** | 0 | 0.1 | ns |

Number in parentheses indicate cohort sizes. "Rapid up" and "Rapid down" cohorts were obtained by filtering for a > 2-fold change at the 4-hour time point, combined with a consistent expression trend at the remaining time points. "Up" and "Down" cohorts defined by requiring at least 3 out of 4 time points to be either up- or downregulated. Bold: *p*-value < 10⁻⁶.
*O* observed overlap, *E* expected overlap, *P* two-sided *p*-value (Chi square test), *TM* transmembrane, *WB* whole body, *ns* not significant, *n/a* not applicable.

genes such as *Drip*, *GstD2* and *rdog*. *Fer1HCH* and *Fer2LCH* were both part of cluster #7 (22 genes), which showed peak expression at the 12-h time point. This cluster also included Copper transporter 1B (*Ctr1B*).

**Analysis of whole-body samples**
In total, we identified 176 genes in WB samples that responded significantly to changes in dietary iron. Of these, 55 genes overlapped with the 486 genes from the gut cohort, including *fire*, *fire-like* and *firewood* (Fig. 1D). The 176 WB cohort was also enriched for iron/metal genes (Table 2). Cluster analysis grouped the *fire* gene complex and *Malvolio* again into a single group (cluster #1, 8 genes), just like we saw in the gut cluster #1. Overall, a total of six genes were common to both clusters, since two additional genes, *CG5892*, which encodes a putative acyl transferase and *CG18179*, encoding a predicted serine protease, were found in either cluster, despite being the result of completely independent experiments (Fig. 2B, Supplementary Data 4).

A set of three metallothionin genes, *MtnE*, *MtnD* and *MtnA* belong to genes uniquely affected in the WB cohort that showed upregulation on high iron diets. *MtnE* and *MtnD* were found in WB cluster #4, which was otherwise very similar to the early-responding gut cluster #6, since it also harbored *Zip99C*, *Zip89B*, *galla-1*, *Drip*, *GstD2*. By contrast, *MtnA* clustered with *Fer1HCH*, *Fer2LCH* and *Ctr1B* (cluster #6, 19 genes), which was overall similar to the late responders in gut cluster #7, whereas *MtnB* appeared to be gut-specific. These findings suggest tissue compartmentalization among metallothioneins, combined with coordinated temporal regulation. Also specific to the non-gut WB cohort was *AGBE*, a key regulator of Iron regulatory protein 1A, critical for maintaining cellular iron homeostasis[41].

**Analysis of BRGC samples**
We identified a total of 88 DEGs in the BRGC samples, which showed little overlap with WB and gut samples (Fig. 1D) and could be divided into three clusters (Supplementary Fig. S2). Cluster #1 genes showed strong upregulation in response to iron and this cluster was significantly enriched for seven heat shock protein genes (Supplementary Fig. S2A, B, Table 2). PG-specific RNAi targeting these heat shock genes revealed moderate developmental delays that could be rescued by iron supplementation, and in the case of Hsp70Bbb, high lethality (Supplementary Fig. S3A). A plausible link between iron and heat-shock proteins is provided by Hsp70, a known mediator of Fe-S cluster transfer to client proteins[63–65], which physically interacts with Hsp22 as shown in this study (Supplementary Fig. S3B). Co-immunoprecipitation followed by MS also revealed that Hsp22 interacted physically with RSeFP, a mitochondrial Rieske iron-sulfur protein, mitochondrial Aconitase, and Fer1HCH, strongly suggesting that Hsp22 plays a role in iron metabolism (Supplementary Fig. S3B, C, Supplementary Data 5). Five of the heat shock genes showed strong transcript enrichment (Supplementary Fig. S3D) in the ring gland, consistent with the notion that the PG is iron-rich[25,41].

Cluster #2 represented genes with moderate downregulation in response to iron. Cluster #3 was clearly distinct, and represented the biggest cohort. Remarkably, this cluster of 58 genes was characterized by a surge in expression at the 16-h time point under iron-deprived conditions. This surge was absent under iron-rich conditions, raising the possibility that this gene cohort is linked to a checkpoint that assesses iron stores. Intriguingly, the critical weight checkpoint[66], by which larvae assess whether they have sufficient nutrients to survive metamorphosis, is normally just prior to this surge (0–12 h after the L2/L3 molt), suggesting that the induction of cluster #3 genes is caused by a failure to fulfill the critical weight checkpoint. Cluster #3 was enriched for 23 genes encoding proteins with a chitin-binding domain, but only Mco4 had an obvious link to iron, as it is orthologous to yeast Fet3p, a multi-copper oxidase involved in high-affinity iron import[67]. Like all cluster #3 genes, *Mco4* showed significant upregulation upon iron depletion, and a detailed functional analysis of Mco4 will be presented below.

**RNAi against candidate genes revealed iron storage defects**
We carried out two strategies to examine the validity of the RNA-Seq results. First, we selected candidate genes for all three sample types, and carried out qPCR. The results closely matched the RNA-Seq data (Supplementary Fig. S4). Next, we tested whether any of the genes identified in the gut cohorts was associated with phenotypes linked to iron metabolism. To this end, we analyzed 24 RNAi lines by staining dissected guts from *NP3084-GAL4 > DEG-RNAi* with Prussian Blue to track the abundance and spatial distribution of gut-stored ferritin (Fig. 3A, B). The genes were selected from Table 1 and our DeSeq2 analysis (Supplementary Data 2) to include lowly expressed genes that appeared significant in the latter analysis, but were not found by the Arraystar software.

Larvae were reared on 120 μM BPS (iron depletion, no Prussian Blue staining) or 1 mM FAC (iron repletion). In controls, iron feeding resulted in a spatial expansion of the Prussian Blue stain compared to standard fly food, where the stain is limited to the iron region[68]. Excess

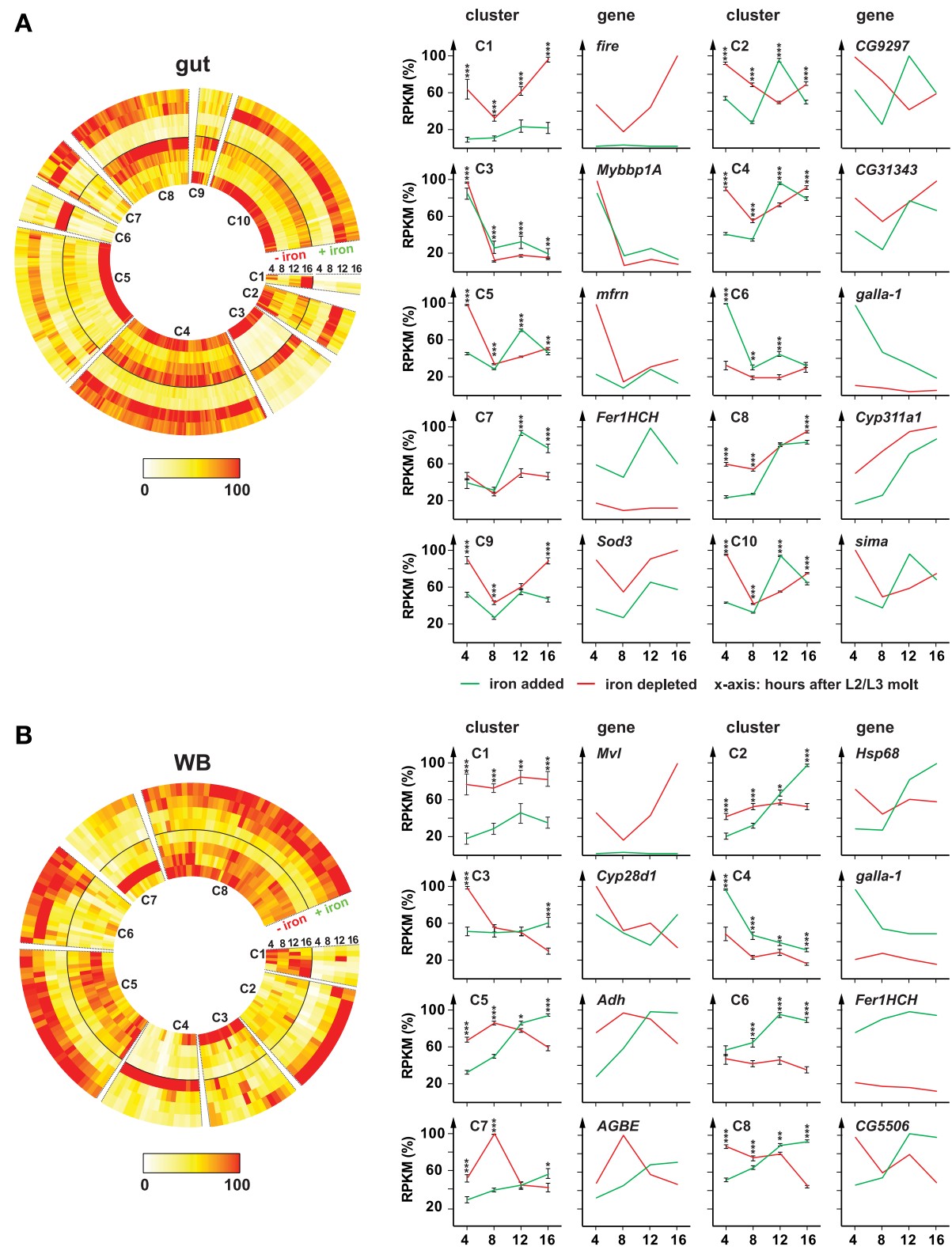

iron absorption is normally met with the spatial expansion of ferritin gene expression in the gut, followed by iron deposition into ferritin. The spatial expansion of iron-loaded ferritin can be affected in multiple ways (e.g., faulty iron transport, disruption of ferritin gene expression, failure to deposit iron into ferritin, blocked iron export from the gut), resulting in both stronger and weaker stains. We observed different types of responses in the 24 RNAi lines tested:

Importantly, half of the RNAi lines, including *firewood*- and *fire*-RNAi, showed a lack of spatial expansion of iron in the anterior midgut, suggesting their function was physiologically relevant in the iron homeostatic response. Five of the RNAi lines showed Prussian Blue patterns and intensities that were indistinguishable from controls (*Desi*, *CG13454*, *CG33270*, *MtnA*, *MtnB* in Fig. 3A, B). RNAi targeting *Zip99C* (aka Zip13), abolished Prussian Blue staining altogether,

**Fig. 2 | Differentially expressed genes in response to dietary iron supplementation in the gut and whole-body samples. A, B** Cluster analysis of 486 and 176 differentially expressed genes in the gut and whole body (WB), respectively. On the left, 10 clusters (C1–10) and 8 clusters (C1–8) are shown as circular heat maps. For each gene, the highest RPKM value was set to 100 (red), with the remaining seven time points normalized to this maximum. Numbers indicate hours after the dietary switch. The inner four circles depict iron-depleted conditions ("−iron"), and the outer four circles show iron-supplemented conditions ("+iron"). The graphs on the right show cluster profiles (C1–10 for gut and C1–8 for WB), each paired with a representative gene. All graphs show normalized RNA-Seq expression data. Time points (x-axis) indicate hours after the dietary switch. For each gene, the highest RPKM value was set to 100%, and the remaining time points were scaled proportionally. Each cluster profile represents the average relative expression (% of maximum) across all genes in that cluster at each time point. Error bars indicate the standard error at each time point. Asterisks denote significance based on a two-sided Student's $t$ test (*$p < 0.05$ and ***$p < 0.001$). Source data are provided in the accompanying source data file.

consistent with its role in loading iron into ferritin[52]. A strong reduction in the Prussian Blue staining in the iron region was also observed when *ABCA* or *Ugt86Dc* or *Drip* were targeted. By contrast, six lines displayed a stronger stain in and around the iron region, of which *Zip89B*- and *CG14626*-RNAi showed the additional expansion in the anterior midgut, whereas *Fer1HCH*-, *Fer2LCH*-, *Hsp68*- and *CG18179* were defective in the latter response (Fig. 3A, B). Finally, RNAi targeting *galla-1* showed further expansion of the Prussian Blue staining into the posterior midgut, albeit at reduced strength, reminiscent of observations of iron accumulation in mutants of iron-sulfur cluster assembly factors[69]. Interestingly, six of the genes tested here were part of gut cluster #6 (*CG7763, Drip, galla-1, Zip99C, Zip89B, MtnB*), however, their corresponding Prussian Blue patterns fell into five classes. As such, coordinated regulation does not imply similar function, but rather participation is the same molecular response.

In summary, Prussian Blue staining is a sensitive tool to assess whether a given RNAi line interferes with storing excess iron in the gut, and allowed us to quickly corroborate that candidate genes from the RNA-Seq data are involved in the regulation of iron physiology.

### The *fire* gene complex and *firewood*: Two ferric reductases and a cytochrome b5 electron donor function in *Drosophila* iron absorption

The RNA-Seq data showed that the most strongly affected genes in the gut samples were *fire*, *fire-like* and *firewood* (Table 1), which, together with *Malvolio*, riboflavin transporter *Rift*, acyl transferase *CG5892*, and peptidases *CG18179* and *CG9416* formed a small cluster of genes that showed a rapid (after 4 h) downregulation in response to iron starvation/re-feeding (Fig. 2A, Supplementary Data 4). Malvolio/DMT1 transports $Fe^{2+}$, which is produced from dietary $Fe^{3+}$ via a reduction reaction. Since Fire and Fire-like represent CYB561 family members, these enzymes are likely reducing $Fe^{3+}$ to $Fe^{2+}$ in the *Drosophila* gut, while their co-regulation with Firewood suggested the latter acts as an electron donor (Fig. 4A). To test this directly, we monitored the formation of BPS-$Fe^{2+}$ precipitates in the gut. Since the chelation of BPS with $Fe^{2+}$ results in a red precipitate, it can be used to detect high ferric reductase activity in the gut (Fig. 4B)[14]. Interestingly, BPS addition resulted in a narrow red stain anterior to the iron region (named "BPR" for "BPS precipitation region", see Fig. 4B). This stain was abolished in *fire*-RNAi, and strongly reduced in *firewood*-RNAi guts, but remained unaffected in *fire-like*-RNAi larvae (Fig. 4C and Supplementary Fig. S6A), indicating that both Fire and Firewood contribute to dietary iron reduction. Consistent with this, the BPR was absent in both the *fire$^{2xKO}$* double null mutant (*fire$^{-/-}$*, *fire-like$^{-/-}$*, Fig. 4C and Supplementary Fig. S7A) and the *firewood$^{KO}$* single mutant, which we generated by CRISPR/Cas9-mediated deletion of the *fire/fire-like* and *firewood* loci (Supplementary Figs. S5A, B, and S6A, B). In *fire$^{2xKO}$* heterozygotes, however, the BPR was present, suggesting that the loss of BPS staining in homozygotes was indeed due to the absence of functional Fire and Fire-like proteins (Supplementary Fig. S7A).

We next asked whether the BPR aligned with the spatial expression of *fire* and *fire-like*. To our surprise, RNA in situ hybridization revealed that both transcripts were expressed anterior to the BPR, with a noticeable drop in expression in the BPR itself (Fig. 4D). *Fire* also showed significant expression in the iron region ("IR", see Fig. 4B), which increased substantially in BPS-supplemented food, consistent with our finding that Prussian Blue expansion fails to occur in *fire*-RNAi animals (Figs. 3 and 4D). We further assessed the expression of the *firewood* and *Mvl* transcripts to examine whether they displayed similar spatial distribution in the gut (Fig. 4D). Both *firewood* and *Mvl* transcripts were detected in the anterior midgut and the IR, comparable to *fire* and *fire-like*. The results were highly consistent with the cluster analysis that placed these four genes in a single cluster (Supplementary Data 4).

To assess subcellular localization, we generated a double knock-in CRISPR line in which the *fire* and *fire-like* loci were replaced with N-terminally tagged *fire-mCherry* and *fire-like-eGFP* alleles (Supplementary Fig. S8A). In agreement with the RNA in situ data, both proteins were detected in the anterior midgut (upstream of the BPR), the iron region, and the posterior midgut (Fig. 4E, Supplementary Fig. S8B, D).

Under iron-depleted conditions, Fire localized predominantly to the apical membrane of anterior and posterior midgut cells (Fig. 4E and Supplementary Fig. S8D), facing the gut lumen. In the iron region, however, the high expression levels made *fire* appear ubiquitously distributed throughout the cells (Fig. 4E and Supplementary Fig. S8E). Fire-like exhibited a similar but weaker expression pattern, with strongest localization in the anterior midgut, reduced presence in the posterior midgut, and no detectable signal in the iron region (Fig. 4E, Supplementary Fig. S8B, D), consistent with the in situ hybridization results. We reasoned that the discrepancy between *fire/fire-like* expression and the BPR was rooted in pH differences in the gut, and that the BPS-$Fe^{2+}$ complex precipitates only in low pH[70,71]. To test this, we monitored the pH in the gut, for which we fed animals with bromocresol purple sodium dye designed to visualize regions of different acidities, while maintaining the BPS stain in the red BPR region. This approach revealed that the BPR is located in a small acidic region just downstream of the end of the neutral anterior midgut segment (Fig. 4F), raising the idea that the stain detected by BPS had precipitated locally after transitioning from a neutral to an acidic section of the gut. If true, the reduction of ferric to ferrous iron likely occurs in the anterior midgut where we detected *fire* and *fire-like* expression, but does not become visible until precipitation in acidic conditions.

To test whether the BPS stain was indeed dependent on acidic environments, we fed flies acetazolamide, a known inhibitor of carbonic anhydrase, an enzyme that drives luminal acidification[72]. As expected, inhibiting acidic conditions in the larval gut also abolished the BPS stain (Fig. 4F, G), consistent with the notion that the BPS stains in the BPR are due to soluble BPS-$Fe^{2+}$ complexes entering an acidic environment, upon which they precipitate and become visible. Finally, we asked whether we could re-establish the BPS stain in the BPR region in *fire$^{2xKO}$* animals by adding ascorbate to the medium. The idea was that excess ascorbate would reduce dietary $Fe^{3+}$ to $Fe^{2+}$, thereby increasing the concentration of BPS-$Fe^{2+}$ complexes entering the gut, which would then precipitate in the BPR. Indeed, adding ascorbate resulted in a detectable BPS stain even in the absence of both Fire and Fire-like (Fig. 4C and Supplementary Fig. S7A), confirming that the BPR does not overlap with the expression pattern of these genes, but

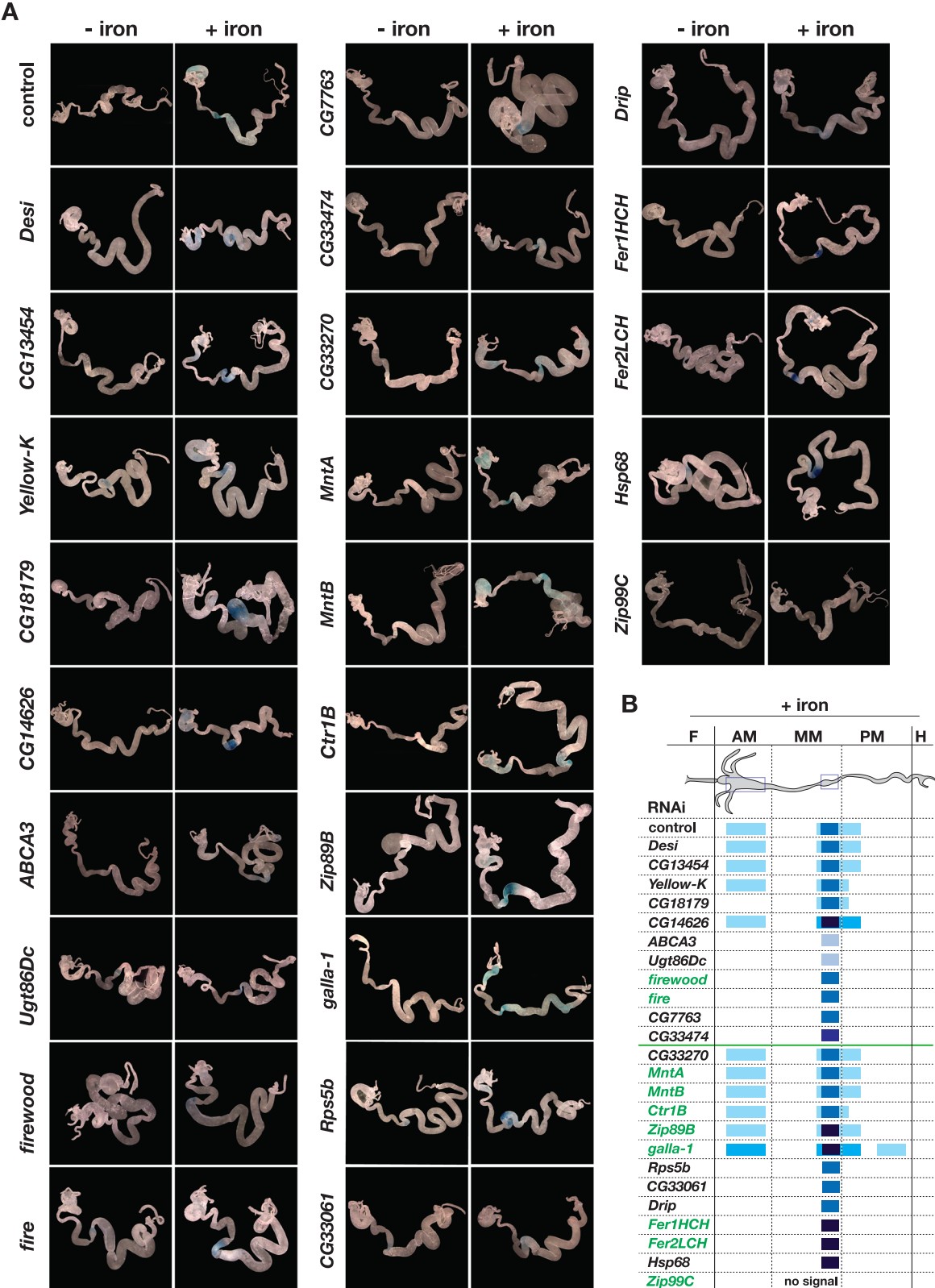

**Fig. 3 | Prussian blue staining of larval guts from selected RNAi lines. A** For each condition, 5–10 larval guts were stained with Prussian blue (ferrocyanide) and representative images are shown. Larvae were reared on iron-depleted ("− iron", BPS-supplemented) or iron-supplemented ("+ iron") diets. RNAi was driven by *NP3084-Gal4*, and *NP3084 > w^{1118}* animals served as controls. Blue staining indicates both the spatial distribution and relative levels of ferric iron accumulation. **B** Schematic summarizing the Prussian blue staining results shown in panel (**A**). Genes above the green line show higher expression under iron-depleted conditions, while those below the line are more highly expressed under iron-supplemented conditions. Gene names in green indicate known links to iron biology. AM anterior midgut, MM middle midgut, PM posterior midgut. Source data are provided in the accompanying source data file.

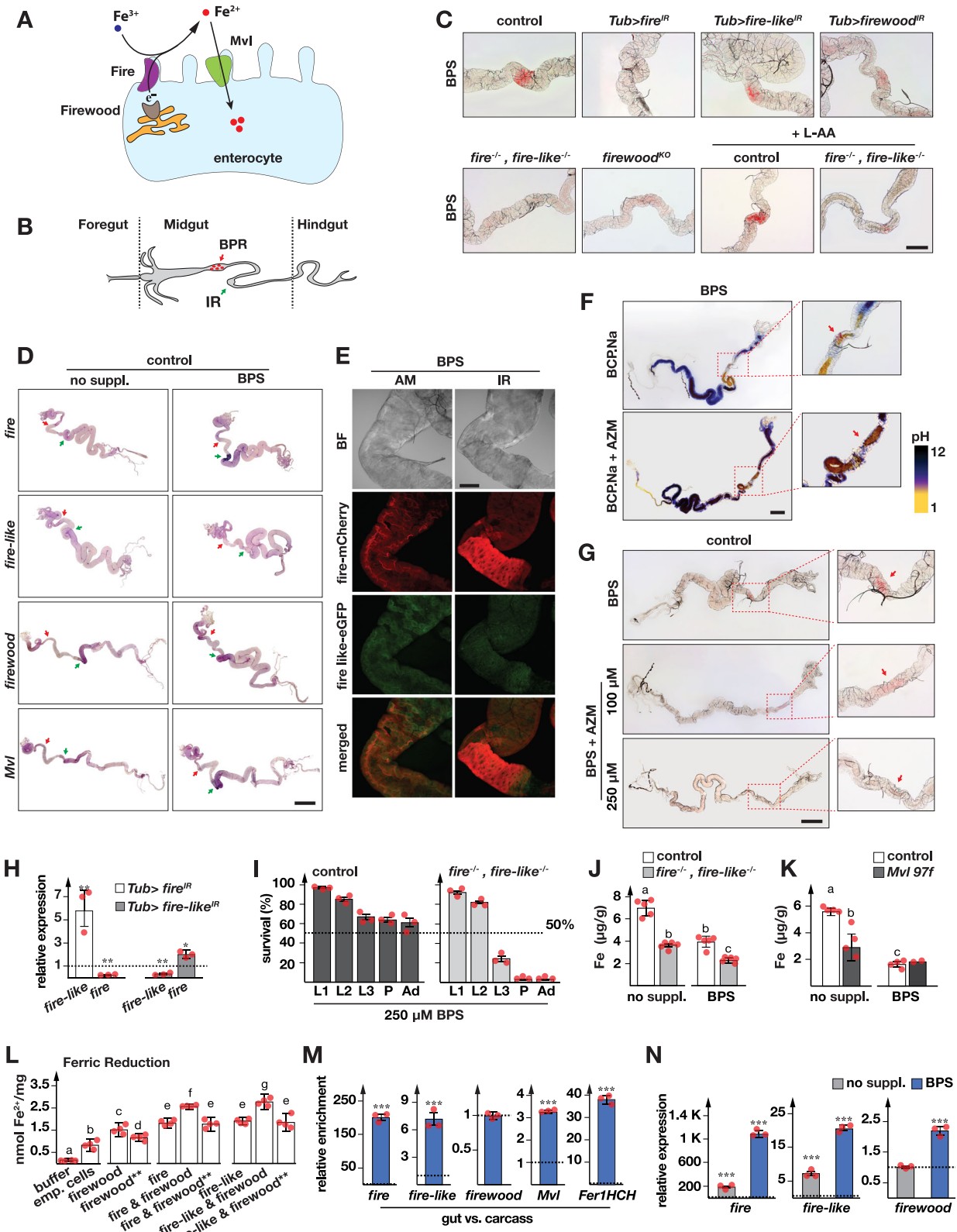

instead marks a downstream region where a change in acidity triggers local precipitation of BPS-Fe$^{2+}$ complexes.

Ubiquitous RNAi targeting *fire-* or *fire-like* had no effect on viability and morphology on regular fly food. Both genes are adjacent to each other, suggesting they are the result of a gene duplication, and may overlap in their functions. Indeed, when we knocked down *fire*, the *fire-like* gene was induced 5.5-fold, and RNAi against *fire-like* caused a 2-fold

upregulation of *fire*, indicating compensatory regulation (Fig. 4H). The double-mutant *fire*$^{2xKO}$ displayed no apparent phenotypes on regular fly food and food supplemented with moderate BPS concentrations (120 μM), suggesting that the presence of reducing agents in the fly media (e.g., propionic acid, which is necessary to prevent bacterial growth) provide sufficient ferrous iron for survival. When *fire*$^{2xKO}$ animals were reared on a diet containing 250 μM BPS to chelate ferrous

**Fig. 4 | Ferric iron reduction mediated by Fire, Fire-like and Firewood.**
**A** Schematic of iron reduction and uptake mediated by Fire, Fire-like, Firewood and Malvolio (Mvl). **B** Diagram of the larval midgut showing the iron region (IR, green arrow) and BPS precipitation region (BPR, red arrow) in third instar larvae (L3). **C** Brightfield images of the BPR in midgut sections of control ($Tub > w^{1118}$), $Tub>fire^{IR}$, $Tub>fire-like^{IR}$, $Tub>firewood^{IR}$, $fire^{-/-}$, $fire-like^{-/-}$ ($fire^{2xKO}$) and $firewood^{KO}$ animals reared on BPS-supplemented media. Also shown: control ($w^{1118}$) and $fire^{2xKO}$ larvae reared on BPS- and L-ascorbic acid (L-AA)-supplemented media. Scale bars: 80 μm. **D** In situ hybridization brightfield images of $fire$, $fire-like$, $firewood$ and $Mvl$ in control guts ($Tub > w^{1118}$). Larvae were reared on normal (no suppl.) or BPS-containing diets, and dissected at 44 h after the L2/L3 molt. Red arrows: BPR. Green arrows: IR. Scale bars: 150 μm. **E** Fluorescent images of the anterior midgut (AM) and IR region in larvae expressing $fire-mCherry$, $fire-like-eGFP$, reared on a BPS-supplemented diet. Scale bar: 80 μm. BF Brightfield, Green: Fire-like-eGFP, Red: Fire-mCherry. **F** BPR and gut pH. Top: controls ($w^{1118}$) fed with BPS and bromocresol purple sodium (BCP.Na). Bottom: as above, with acetazolamide (AZM) added. Scale bars: 150 μm. **G** Brightfield images of the BPR in control larvae ($w^{1118}$) fed i) BPS, ii) BPS + 100 μM AZM or iii) BPS + 250 μM AZM. Scale bars: 150 μm. **H** qPCR analysis of

$fire$ and $fire-like$ in L3 guts from $Tub>fire^{IR}$ and $Tub>fire-like^{IR}$ larvae. **I** Survival of control and $fire^{2xKO}$ on 250 μM BPS media. Dotted line: 50% pupariation. L1/L2/L3: 1st, 2nd and 3rd instar larvae. P: Pupae. Ad: Adults. Error bars: standard error ($n$ = 3). **J, K** Total iron levels in L2 larvae (20 h after the L1/L2 molt) of control ($w^{1118}$), $fire^{2xKO}$ and $Mvl^{97f}$ larvae reared on normal or 250 μM BPS-supplemented diets. Iron was quantified by ICP-OES and normalized to body weight. Error bars: standard deviation ($n$ = 5 for **J**; $n$ = 4 for **K**, except $Mvl^{97f}$ on BPS: $n$ = 2). **L** Ferric reductase activity of lysis buffer ("buffer") and cell lysates expressing empty vector ("emp. cells"), wild-type $firewood$ ("$firewood$"), double-mutant $firewood$ ("$firewood^{**}$"), $fire$ and $fire-like$, or combinations thereof. $Fe^{2+}$ levels were measured by ferrozine assay. Error bars: Standard error ($n$ = 4). Statistics in (**J–L**): ANOVA; different letters indicate significant differences (two-sided $p < 0.05$). **M** qPCR analysis of $fire$, $fire-like$, $firewood$, $Mvl$ and $Fer1HCH$ in gut vs. carcass (larval body without gut). **N** Expression of $fire$, $fire-like$ and $firewood$ on normal (no suppl.) or BPS-supplemented diets. **H, M, N** Data represent three biological replicates, each tested in triplicate. Error bars: 95% confidence intervals, center lines indicate means. Asterisks: significance by two-sided Student's $t$ test (***$p < 0.01$). Source data are provided in the accompanying source data file.

iron before gut entry, less than 5% reached the pupal or adult stage, compared to ~60% survival in controls (Fig. 4I). This severe reduction in survival correlated with a marked decrease in larval iron levels, measured by Inductively Coupled Plasma Optical Emission Spectrometry (ICP-OES). $fire^{2xKO}$ mutants contained ~55% less iron than controls, both on regular food and food supplemented with 250 μM BPS (Fig. 4J). This degree of iron depletion was comparable to that observed in $Mvl^{97f}$ mutants[73] on regular food (~50% reduction) in iron, but unlike $fire^{2xKO}$, $Mvl^{97f}$ mutants reared on BPS media retained iron levels similar to $w^{1118}$ controls (Fig. 4K).

To directly confirm the ferric reductase activity of Fire and Fire-like, we conducted an ex vivo colorimetric assay in S2 cells using ferrozine as a $Fe^{2+}$-specific chelator[74,75] (Fig. 4L). Expression of $fire$ or $fire-like$ in S2 cells resulted in ~2.5-fold increase in ferric reduction compared to cells transfected with the empty vector (Fig. 4L). To assess whether Firewood contributes to ferric reduction via electron transfer, we co-transfected S2 cells with either wild-type $firewood$ ($firewood^{WT}$) or a mutant version ($firewood^*$), in which the conserved heme-binding histidine residues (H39 and H63) were mutated to alanine to disrupt electron transfer (Supplementary Fig. S6C, D), together with $fire$ and $fire-like$. Ferric reductase activity of Fire and Fire-like increased by ~1.5-fold in the presence of Firewood$^{WT}$, but this enhancement was absent when mutant Firewood$^{**}$ was used (Fig. 4L).

A final aspect we examined was whether $fire$ gene expression was gut-specific. To this end, we compared gut samples with all remaining tissues ("carcass"). Overall, $fire$ transcripts showed 200-fold enrichment (Fig. 4M), whereas $fire-like$ transcripts were 7-fold enriched in the gut. By contrast, $firewood$ expression was comparable to carcass. For reference, $Malvolio$ and $Fer1HCH$ showed ~3-fold and 37-fold higher expression levels in the gut (Fig. 4M). On BPS-containing media, $fire$ was further upregulated to ~1100-fold, $fire-like$ to ~21-fold, whereas $firewood$ roughly doubled its expression (Fig. 4N). We conclude that $fire$ expression is gut-specific, and encodes the main gut ferric reductase in the fly. Two additional CYB561 genes are located near the $fire/fire-like$ (FFL) locus, which raised the possibility that they work in a redundant fashion with $fire$ and $fire-like$. One, $CG10337$, lies 18.5 kb upstream of the FFL locus (separated by two other genes), whereas $CG10165$ lies 37.0 kb upstream of the FFL locus (separated by another five genes from $CG10337$). When we examined the transcript levels of $CG10337$ and $CG10165$ and the other four CYB561 family members, we observed ~5-fold, ~11-fold and 3.5-fold transcript enrichment in the gut for $CG10337$, $CG10165$ and $nemy$, respectively (Supplementary Fig. S5C). The $CG8399$ and $CG3592$ CYB561 transcripts did not exhibit enrichment in the gut. While the close chromosomal vicinity of the four CYB561 genes could suggest overlapping functions in vivo, we did not observe a corresponding response to changes in dietary iron in the

RNA-Seq data, with $CG10165$ showing around 1.1-fold differences throughout the four time points, whereas $CG10377$ registered 1.25- to 2.1-fold changes. Similarly, the remaining four CYB561 genes ($CG3592$, $nemy$, $CG1275$ and $CG8399$) all showed changes lower than ~2-fold. This was validated via qPCR, which showed that iron starvation most strongly affected $fire$ and $fire-like$, with little effect on the other members, apart from $CG10337$, which exhibited a 3.5-fold upregulation in the gut (Supplementary Fig. S5C). By contrast, $fire$ and $fire-like$ showed maximal fold changes of 78- and 52-fold (Table 1), indicating that the response to dietary changes in iron concentrations was selective for the $fire$ and $fire-like$ genes. This is supported by the $fire$-RNAi and $fire^{2xKO}$ deletion studies in conjunction with BPS feeding, which demonstrated that the formation of BPS/$Fe^{2+}$ precipitates fail to occur when Fire function is impaired or absent. This strongly suggests that, apart from Fire and Fire-like, no other CYB561 proteins participate in reducing dietary ferric iron in the gut, despite the fact Nemy and CG1275 display higher sequence similarities to human DCYTB (Supplementary Fig. S5D, E).

## Mco4 functions in cellular iron import

We initially identified $Mco4$ due to its upregulation in response to BPS at the 16-h time point in the BRGC RNA-Seq data, which showed a ~2000-fold increase compared to the same time point under iron-replete conditions (Fig. 5A and Supplementary Fig. S2). In the 4-, 8- and 12-h time points, $Mco4$ was nearly undetectable in BRGC samples (Supplementary Fig. S2), whereas absolute $Mco4$ transcript at 16-h in the BRGC/BPS cohort were comparable to $Mco4$ levels in the gut under all conditions. After one generation on 120 μM BPS media (representing moderate iron depletion), qPCR revealed 2-, 5- and 13-fold upregulation of $Mco4$ in RG samples (Fig. 5B), confirming that $Mco4$ is transcriptionally induced in response to iron deprivation. When we analyzed Mco4 transcript levels in the gut vs. carcass, we saw a 6000-fold enrichment in the gut, suggesting Mco4 exhibits gut-specific expression under iron-replete conditions (Fig. 5C). $Mco4$ was not picked up in the gut RNA-Seq data, since $Mco4$ expression levels were fairly robust at all time points, regardless of iron status, a finding we confirmed via qPCR, which showed a moderate ~1.6-fold upregulation of $Mco4$ under BPS in whole gut samples (Fig. 5C). By contrast, we saw strong $Mco4$ upregulation in the carcass in response to iron depletion, raising the possibility that the ring gland is not the only tissue that exhibits $Mco4$ upregulation in the presence of BPS (Fig. 5C).

Next, we carried out in situ hybridization to analyze $Mco4$ expression in the gut (Fig. 5D). Surprisingly, we found strong $Mco4$ expression in the proventriculus (PV), where the spatial distribution formed a torus-like structure. Under iron depletion via BPS, $Mco4$ expression increased moderately throughout the main sections of the

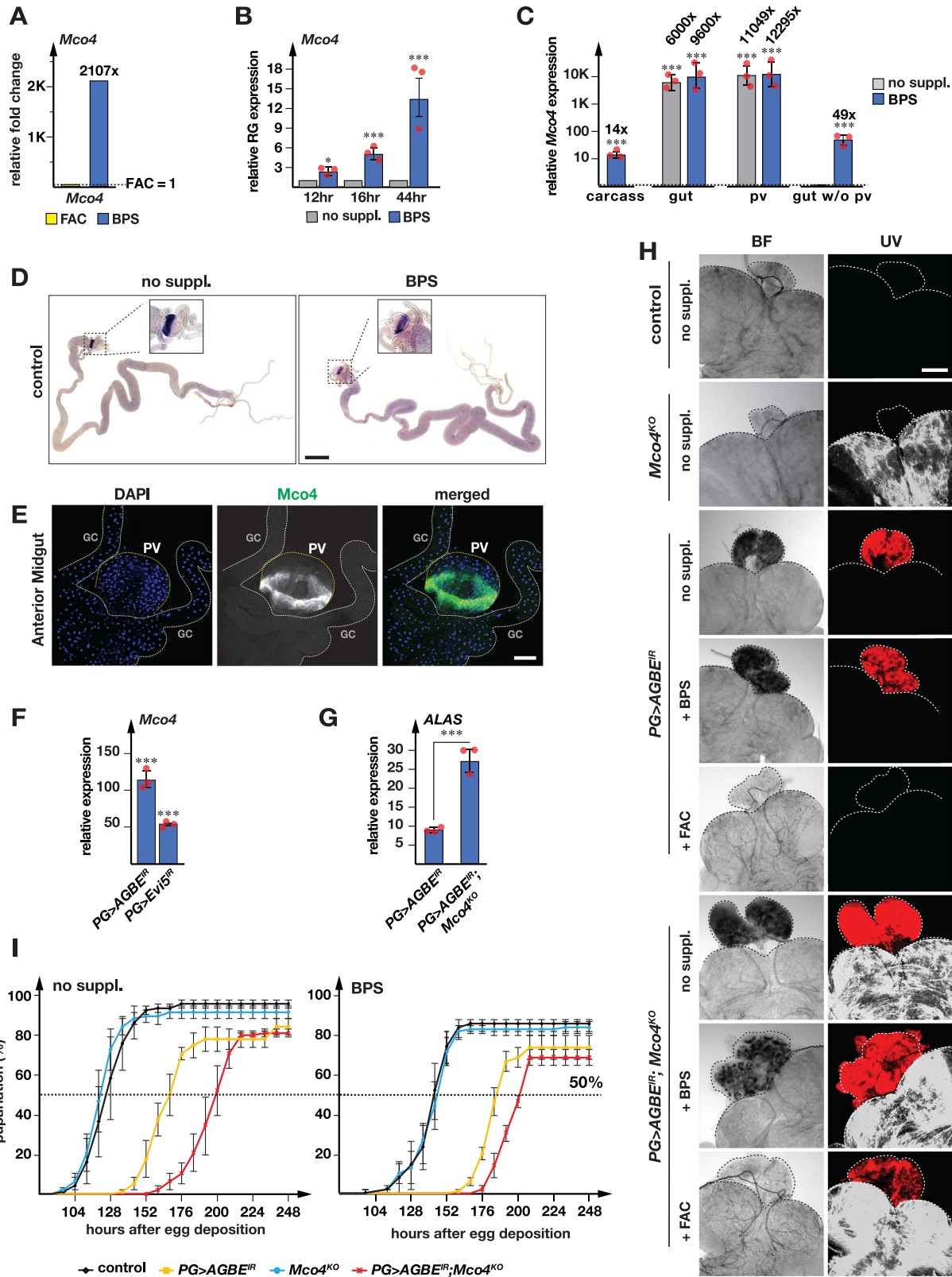

midgut, but appeared stable in the PV (Fig. 5D). This was confirmed through qPCR analysis of *Mco4* expression in the PV and in guts where we had removed the PV. *Mco4* expression was virtually unchanged in the PV, but we observed a nearly 50-fold induction in the rest of the gut (Fig. 5C). We then generated a 3x-Flag-tagged *Mco4* knock-in via CRISPR/Cas9, which allowed us to confirm the torus-like expression pattern in the PV (Fig. 5E).

Given that *Mco4* is orthologous to yeast *Fet3*, we wondered whether Mco4 was part of a high-affinity iron import complex in *Drosophila*, which would represent the first report of such a system in an animal. Since we saw significant upregulation of *Mco4* in the ring gland in response to iron depletion, we worked first in this tissue and examined whether Mco4 would be also induced if we disrupted iron homeostasis by genetic means. To this end, we analyzed *Mco4*

**Fig. 5 | Mco4 is required for iron uptake into the prothoracic gland. A** RNA-Seq analysis of *Mco4* expression in G6 BRGCs from larvae reared on BPS- or FAC-supplemented (iron-rich, normalized to 1) diets. **B** qPCR analysis of *Mco4* in ring glands (RG) of control (*w[1118]*) larvae at 12, 16 and 44 h after L2/L3 molt. Larvae were reared on normal (no suppl.) and on BPS- (iron-depleted) diets. **C** qPCR analysis of *Mco4* expression in the carcass, gut, proventriculus (PV), and gut without proventriculus (gut w/o PV) of control (*w[1118]*) larvae reared on normal (no suppl.) or BPS-supplemented diets. In (**B**, **C**), expression data are based on three biological replicates, each tested in triplicate. Error bars represent 95% confidence intervals, asterisks denote significance by two-sided Student's *t* test (***p < 0.01). **D** Brightfield images of *Mco4* in situ hybridization in control (*w[1118]*) guts from larvae reared on normal or BPS-supplemented diets. Scale bars: 150 μm. **E** Immunodetection of Mco4 in the proventriculus (PV) of *Mco4[3xFLA]* knock-in larvae. Blue: nuclei (DAPI), gray and green: Mco4 signal. Scale bars: 100 μm. **F** qPCR analysis of *Mco4*

expression in ring glands (RG) from *PG > AGBE[IR]* and *PG>Evi5[IR]* larvae, normalized to *PG > w[1118]* controls (set to 1). **G** qPCR analysis of *ALAS* expression in ring glands from *PG > AGBE[IR]* and *PG > AGBE[IR]; Mco4[KO]* larvae. **F, G** Data are from three biological replicates, each tested in triplicate. Error bars: 95% confidence intervals; asterisks: two-sided Student's *t* test (***p < 0.01). **D–G** All larvae were dissected at 44 h after the L2/L3 molt. **H** Red autofluorescence in ring glands of i) control, ii) *Mco4[KO]*, iii) *PG > AGBE[IR]* and iv) *PG > AGBE[IR]; Mco4[KO]* larvae reared on normal (no suppl.), iron-rich (+FAC) and iron-depleted (+BPS) diets. UV ultraviolet, BF brightfield. Scale bars: 250 μm. **I** Developmental timing analysis of control, *Mco4[KO]*, *PG > AGBE[IR]* and *PG > AGBE[IR]; Mco4[KO]* larvae reared on normal (no suppl.) or BPS-supplemented diets. Y-axis: percentage pupariated. X-axis: hours after egg deposition. Dotted line marks 50% pupariation. Error bars: standard error (*n* = 3); center lines in (**B**, **C**, **F**, **G**, and **I**) represent means. Source data are provided in the accompanying source data file.

expression in genetic backgrounds where we disrupted either *AGBE* or *Evi5* in a PG-specific manner, which disrupted iron homeostasis and iron transport respectively[25,41]. Remarkably, we observed a ~100-fold induction of *Mco4* in *PG > AGBE*-RNAi animals, and a 50-fold induction in *PG>Evi5*-RNAi samples (Fig. 5F). This indicated that *Mco4* was not expressed under iron-replete conditions, but was strongly induced once iron became scarce, either by nutritional or genetic means.

Given the strong transcriptional induction of *Mco4* in both *AGBE*- and *Evi5*-RNAi lines, we wondered whether removing Mco4 in either of these backgrounds would exacerbate iron phenotypes in the PG. To this end, we generated an *Mco4* null allele by replacing the endogenous gene with a *3xP3-DsRed* (*Mco4[KO]*) reporter, which causes red fluorescence in the brain, but not the ring gland. *PG > AGBE*-RNAi animals have impaired cellular iron homeostasis, which causes a disruption of heme biosynthesis in the PG[41]. Disrupting heme production is characterized by the upregulation of *Alas*, which encodes the rate-limiting enzyme in heme biosynthesis, as well as heme precursor accumulation, which causes red autofluorescence in the affected tissue. Consistent with our earlier findings, *Alas* was ~8-fold upregulated in *PG > AGBE*-RNAi ring glands. Remarkably, *PG > AGBE*-RNAi; *Mco4[KO]* animals displayed much stronger *ALAS* upregulation (>25-fold, Fig. 5G), demonstrating that removing *Mco4* in *PG > AGBE*-RNAi animals aggravated the impairment of heme biosynthesis due to lower iron uptake. To corroborate the *Alas* results, we next examined autofluorescence levels in these animals.

*PG > AGBE*-RNAi ring glands show strong red autofluorescence due to the disruption of iron homeostasis in this tissue. By contrast, ring glands isolated from *Mco4[KO]* mutants display no auto-fluorescence (Fig. 5H). The red autofluorescence seen in *PG > AGBE*-RNAi ring glands can be completely rescued by supplementing fly media with iron (Fig. 5H). Importantly, this rescue requires Mco4, since *PG > AGBE*-RNAi; *Mco4* animals still exhibit strong red auto-fluorescence despite iron supplementation (Fig. 5H). These genetic data suggest that Mco4 contributes to maintaining cellular iron availability in the PG and are consistent with a role in facilitating iron import. The contribution of Mco4 to *AGBE*-RNAi phenotypes can be also observed on the population level. With standard food or mild BPS concentrations (120 μM), larvae where we removed *Mco4* from an otherwise wild-type background showed the same growth curves as controls and were equally viable (Fig. 5I). By contrast, *PG > AGBE*-RNAi; *Mco4[KO]* animals were developmentally delayed by ~30 h compared to *PG > AGBE*-RNAi larvae, indicating that the transcriptional upregulation of *Mco4* in *AGBE*-RNAi animals is functionally important and likely a compensatory measure to counter low iron concentrations in the PG.

Taken together, these data indicate that *Mco4* is transcriptionally upregulated in response to a drop in cellular iron availability, and that Mco4 is involved in cellular mechanisms that promote iron import into the PG.

## Mco4 is the Fet3p ortholog essential for iron uptake under iron-starvation

Under moderate iron depletion (160 μM BPS), we observed that *Mco4[KO]* mutants displayed only 20% survival in the third generation compared to 80% in controls (Fig. 6A). The inability to survive on moderately iron-depleted media suggested that Mco4 played a role in the absorption of dietary iron in the gut. To test this idea, we determined iron concentrations in whole larvae using Inductively Coupled Plasma Mass Spectrometry (ICP-MS), which demonstrated that *Mco4[KO]* mutants had comparable iron levels to controls when reared on regular fly food, but had ~60% less iron when reared on BPS-supplemented food (Fig. 6B). Thus, *Mco4* is required for iron absorption when bioavailability is low.

Multicopper oxidases can function as laccases, ascorbate oxidases and ferroxidases[76]. To test whether Mco4 had ferroxidase activity, we transfected Schneider 2 (S2) cells with plasmids encoding Hephaestin, a known human ferroxidase, or Mco4, and measured cell extracts in a ferroxidase enzyme assay. Both Hephaestin and Mco4 had comparable ferroxidase activity in this assay (Fig. 6C), consistent with the idea that Mco4 acts as a ferroxidase like its yeast counterpart Fet3p. Next, we asked whether overexpression of Mco4 was sufficient to increase iron concentrations in animals. While total iron was unaffected, we noticed that the iron region of *Mco4*-over-expressing animals showed a stronger Prussian Blue stain, indicative of higher levels of locally stored iron in the form of ferritin (Fig. 6D). Remarkably, in the presence of BPS, which normally results in the absence of a Prussian Blue stain in the iron region (IR), *Mco4* over-expression retained ferritin in the IR, indicating that *Mco4* over-expression increases stored iron in the IR, possibly due to higher absorption rates from the diet. Next, we examined whether this increase in stored iron would allow *Mco4*-overexpressing animals to survive otherwise lethal concentrations of BPS. This is indeed the case. When we reared control or *tub>Mco4*-cDNA animals on 500 μM BPS, controls were unable to complete larval development and never formed pupae. By contrast, *tub>Mco4*-cDNA animals not only formed pupae (Fig. 6E), but 40% of the population reached adulthood (Fig. 6F). To ensure that the lethality seen at these concentrations was solely due to the iron-chelating property of BPS and not due to non-specific effects of the compound, we repeated the experiment with control animals, but also tested a condition where we added simultaneously BPS and Ferric Ammonium Citrate (FAC), which showed that adding back iron completely rescued the lethality seen with BPS (Fig. 6G). Taken together, these data demonstrate that *Mco4* overexpression is sufficient to overcome iron starvation conditions that are lethal for wild-type, consistent with the idea that Mco4 is part of an iron import system that operates even in the presence of high iron chelator concentrations, suggesting a high-affinity mechanism.

Finally, we tested whether *Drosophila Mco4* could functionally replace yeast *Fet3*. For this experiment, we transfected *Fet3* mutants with empty vector, or plasmids harboring cDNAs for *Fet3* or *Mco4*, and

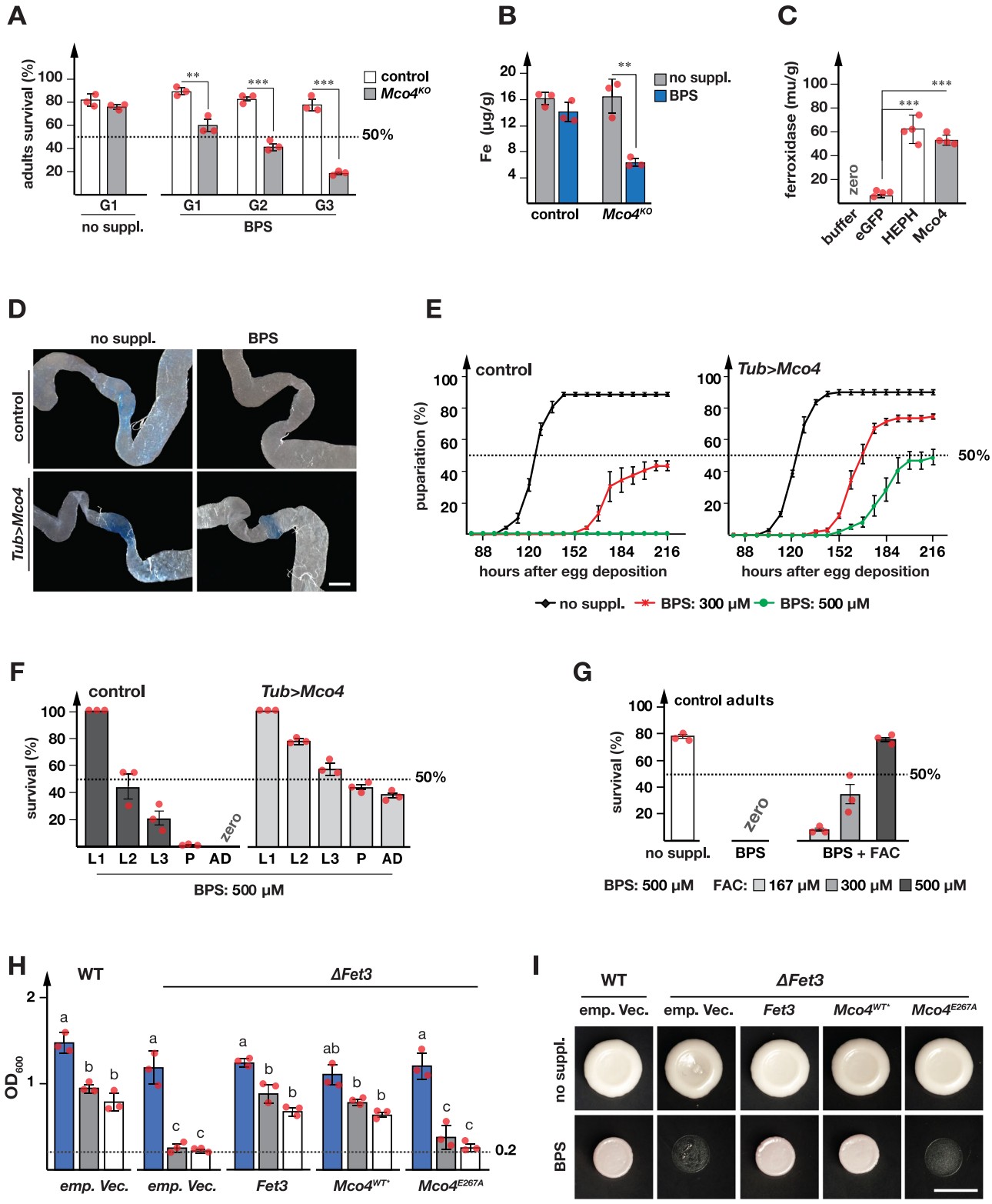

measured growth under normal and iron-depleted conditions. Remarkably, transfection with either *Fet3* or *Mco4* resulted in a comparable rescue of the *Fet3* mutations under iron starvation (Fig. 6H, I and Supplementary Fig. S6B). To investigate whether the rescue of *Fet3* mutants by Mco4 was dependent on the enzyme's ability to bind iron, we generated an Mco4 variant with impaired iron-binding capacity

(Mco4$^{E267A}$) by targeting Glu267 in Mco4 (Supplementary Fig. S7), which corresponds to Glu185 in Fet3p. Glu185 has been shown to play a crucial role in Fe$^{2+}$ binding in Fet3p, electron transfer coupling, or Fe(III) trafficking[77,78]. Transfection of *Fet3* mutants Mco4$^{E267A}$ failed to rescue (Fig. 6H, I and Supplementary Fig. S6B), reinforcing the conclusion that Mco4 has ferroxidase activity, and demonstrating that

**Fig. 6 | Mco4 functions as a ferroxidase in dietary iron absorption. A** Adult survival of *Mco4^KO* and control (*w^1118*) flies on normal (no suppl.) and iron-depleted diets (BPS). Flies were reared for three generations (G1-G3) on BPS-containing media. Dotted line: 50% pupariation. Error bars: standard error (*n* = 3). **B** Iron content of L3 control and *Mco4^KO* larvae measured by ICP-MS. Animals reared on normal (no suppl.) or BPS media, L3 larvae collected at 44 h after the L2/L3 molt. Y-axes: elemental iron (μg) per body weight (gram). Error bars: standard error (*n* = 3). **C** Ferroxidase assay of S2 cell extracts expressing empty vector (emp. vec.), human *hephaestin* (HEPH) or *Mco4*. Error bars: standard error (*n* = 4). Asterisks in (**A–C**) indicate significance (two-sided Student's *t* test, ***p < 0.01). **D** Prussian blue staining of larval guts from *Tub > w^1118* and *Tub>Mco4*-cDNA larvae reared on normal (no suppl.) or BPS-supplemented diets. Blue color indicates ferric iron accumulation. Scale bars: 80 μm. **E** Developmental timing of control (*Tub > w^1118*) and *Tub>Mco4*-cDNA larvae reared on normal diet (no suppl.) or BPS-supplemented food (300 μM, red; 500 μM, green). Y-axis: percentage pupariated. X-axis: hours after egg deposition. **F** Survival analysis of control and *Tub>Mco4*-cDNA larvae on 500 μM BPS-supplemented diet. Dotted line marks 50% pupariation. L1/L2/L3: 1st, 2nd, 3rd instar larvae, P pupae, AD adults. **G** Survival of adult controls (*w^1118*) on i) normal (no suppl.), ii) 500 μM BPS, iii-v) 500 μM BPS plus 160 μM, 300 μM and 500 μM FAC diets. Dotted line denotes 50% pupariation. **H** OD$_{600}$ measurements of wild-type and ΔFet3 yeast transformed with plasmids expressing *Mco4* cDNA, *Fet3* cDNA, or empty vector (emp. vector). Plasmids expressing *Fet3* or *Mco4* also co-expressed *Ftr1*. Cells were grown in synthetic drop-out medium containing zero (no suppl.), 80 and 160 μM BPS. Statistics: ANOVA; different letters denote significant differences (two-sided *p*-value < 0.05). **E–H** Error bars: standard error (*n* = 3); center lines represent means. **I** Growth of transformed yeast colonies on control (no suppl.) and BPS-supplemented (80 μM) media. Scale bars: 5 mm. Source data are provided in the accompanying source data file.

Mco4 can functionally replace Fet3p as part of the yeast high-affinity iron import system.

## Discussion

In this study, we used multi-generational iron starvation to gently sensitize animals towards iron re-feeding, in order to maximize the transcriptional response to a sudden increase in dietary iron levels. Overall, this iron starvation/re-feeding strategy successfully identified known iron metabolism genes, including iron transporters *Malvolio*, *mitoferrin*, *Zip99C* (aka Zip13), ferritin subunits *Fer1HCH*, *Fer2LCH* and a range of metallothionein genes. Importantly, we identified and functionally characterized four genes with no prior roles in insect iron biology: *fire*, *fire-like*, *firewood* and *Mco4*. We demonstrated that these genes represent players that ensure iron absorption under different conditions, identifying a hitherto unknown high-affinity uptake system in animals and a protein electron donor for iron reduction in the DMT1-dependent (low-affinity) iron absorption machinery.

Several findings suggest that Mco4 acts in the gut as a component of a high-affinity iron importer: First, *Mco4* mutants had significantly reduced iron stores and survival when grown on BPS-supplemented media. Second, overexpression of *Mco4* resulted in resistance to extreme levels of BPS, evidenced by the perdurance of iron-loaded ferritin in the gut and the ability to survive under these conditions, whereas controls had no survivors. Third, *Mco4* was upregulated in the PG in response to dietary iron restriction as well as genetic disruption of iron homeostasis and transport. Removing *Mco4* in a *PG > AGBE*-RNAi background abolished the rescue via iron supplementation, demonstrating that the upregulation of *Mco4* is a compensatory measure to import iron directly into the PG. Given that *Mco4* mutants have normal iron levels when reared on regular fly food, we conclude that Mco4 is required for iron import under iron starvation, at least in some peripheral tissues such as the PG.

Our findings in *Drosophila* explain the viability and mild phenotypes of *Malvolio* mutants, and we note that a similar conundrum exists for vertebrate iron homeostasis, where gut-specific loss of DMT1 function led to anemia but not lethality[18,19,79–84]. Similarly, only two of the three mammalian ferroxidases (ceruloplasmin and hephaestin) have been functionally analyzed to date[76], raising the possibility that zyklopen may function in a similar manner to Fet3p and Mco4. Future identification of a permease analogous to yeast Ftr1 will be critical to fully establish whether Mco4 operates as part of a high-affinity iron import system in flies. Such an accessory protein may not be transcriptionally regulated and thus would have escaped detection by our experimental approach.

Under iron-replete conditions, *Mco4* is strongly expressed in the gut. Closer inspection revealed that *Mco4* expression was strongest in the proventriculus (PV), whereas the expression in the midgut epithelium was generally low but strongly upregulated under iron-depleted conditions. The torus-like expression pattern in the PV appeared constitutive and did not respond substantially to iron treatment. Interestingly, the Mco4 torus corresponds to the PV5.5 cell cluster that was recently described in a high-resolution transcriptomics study of the *Drosophila* foregut[85]. In that study, *Mco4*, alongside *CG7567* and *CG5162*, were identified as the major and specific marker genes of this foregut cell type. Neither *CG7567* nor *CG5162* showed iron-dependent regulation in our RNA-Seq data, consistent with the finding that *Mco4* in the PV is not regulated by dietary iron. The role of the PV5.5 cell cluster is currently unclear and requires further investigation. Notably, the PV and the ring gland are connected by neurons, raising the possibility that the PV communicates nutritional status (e.g., iron levels in the diet) to the gland, which in turn may result in systemic hormone signals.

Three of the most strongly downregulated genes in response to shifting animals to a high-iron diet were *fire*, *fire-like* and *firewood*. These three genes are closely linked, as indicated by their predicted functions, significant fold changes, temporal profiles, and their clustering with *Malvolio*. Malvolio/DMT1 transports ferrous iron and other divalent metals into the enterocyte, necessitating the conversion of dietary ferric iron to ferrous iron. Fire and Fire-like represent two of the eight CYB561 proteins encoded by the *Drosophila* genome, however, our data suggest that Fire is the principal gut ferric reductase. CYB561 proteins comprise a family of integral membrane oxidoreductases characterized by two non-covalently bound heme moieties and six transmembrane domains[86,87], and they transfer electrons across the membrane to extracellular substrates such as ferric iron or mono-dehydroascorbate (MDHA). In *Candida albicans*, two proteins, Frp1 and Frp2, function in cellular heme uptake and are related to ferric reductases[88]. Unlike the Fire and Fire-like proteins, however, Frp1/2 do not belong to the CYB561 family. We thus favor the notion that Fire and Fire-like proteins act as canonical ferric reductases. This hypothesis is supported by multiple lines of evidence. First, both Fire and Fire-like exhibited ferric reductase activity in an ex vivo assay. Second, in a wild-type background, the addition of BPS to the diet led to the formation of a distinct precipitate—indicative of BPS-Fe$^{2+}$ complex formation—whereas this response was abolished when *fire* was disrupted. These findings, along with apical membrane localization, suggest that Fire, and to a lesser extent Fire-like, are responsible for generating ferrous iron in the gut, enabling its chelation by BPS and subsequent precipitation.

A fascinating question concerns the source of electrons used by Fire and Fire-like. A commonly accepted idea is that cytoplasmic ascorbate serves as the main electron source for CYB561 enzymes, although redox potential studies suggest that ascorbate cannot account fully for the reduction of CYB561 proteins[86]. In our study, Firewood emerged as a strong candidate for a protein-based electron donor. Firewood contains a cytochrome b5-like heme/steroid-binding domain (Supplementary Fig. S6E, Interpro #IPR001199), a feature shared with 15 other *Drosophila* genes, none of which showed significant differential expression in our RNA-Seq data. Firewood also shows homology to human CYB5b, a known mitochondrial electron

carrier. Firewood was co-regulated with *fire* and *fire-like* in response to iron availability, and promoted their ferric reductase activity in ex vivo assays. Importantly, a mutant version of Firewood, in which heme binding—and thus electron donation—was abolished, failed to enhance the ferric reductase activity of Fire and Fire-like. Furthermore, both *firewood*[KO] mutants and *firewood*-RNAi animals exhibited a marked reduction in the formation of BPS precipitates, indicating impaired ferric reductase function.

In all likelihood, there are additional genes with specific roles in metal and/or iron biology in our RNA-Seq data. For example, the gut cohort was enriched for genes encoding transmembrane transporters (Table 2), some of which have been implicated to directly act in metal detoxification. One such example is the ABC transporter CG10505, first identified in a study where *Drosophila* larvae were fed for 6 h on diets containing either cadmium, zinc or copper[35,89]. Another upregulated ABC transporter in our dataset, encoded by *rdog*, is consistent with an earlier report linking this gene to zinc detoxification[90]. In addition to the iron transporter Zip99C (Zip13), we also identified Zip89B and Zip48C, predicted to act in zinc transport, and the copper transporter Ctr1B, suggesting that there is crosstalk between responses to different metals. We also note the response of *galla-1*, predicted to encode a protein acting in cytosolic iron cluster assembly required for proper IRP-1A and IRP-1B function[41], and also found to be induced by cadmium, zinc or copper in the Schaffner study. Finally, our Prussian Blue analysis revealed abnormal iron ferritin storage in 18 RNAi lines, which suggested reduced iron uptake/transport/storage in these animals.

To conclude, our study has functionally characterized four genes involved in *Drosophila* iron absorption, and has introduced two major concepts: First, that an enzyme-based electron donor (Firewood) delivers electrons for extracellular ferric iron reduction, challenging the notion that cytoplasmic ascorbate is the sole electron source for this process. Second, that Mco4, fly ortholog of yeast Fet3p, is a ferroxidase that acts in high-affinity iron uptake, providing the first example of such a system in animals. Given the evolutionary conservation of many iron metabolism components from flies to humans, these findings pave the way for future studies to explore these novel concepts in the context of human iron metabolism.

## Methods

### Fly stocks
All fly strains used in this study are listed in Supplementary Table S1. The $w^{1118}$ (BL3605) stock was obtained from the Bloomington *Drosophila* stock center and was used for the RNA-sequencing studies in this paper. RNAi lines were obtained from the Vienna stock center and the Bloomington *Drosophila* stock center. We used the following Gal4 drivers: *phm22-Gal4* (a kind gift from Michael O'Connor's lab) and *NP3084-Gal4* (Kyoto *Drosophila* stock center: #113094) to drive GAL4 in the PG or gut cells, respectively. Flies were reared on standard fly food (Nutri-fly, #66-113) at 25 °C and 70% humidity.

### Media preparation and supplementation
Nutri-fly food was prepared according to the guidelines provided by the Bloomington *Drosophila* stock center (Flystuff, #66-113). To alter iron levels we supplemented Nutri-fly food with 100–500 μM Bathophenanthrolinedisulfonic acid disodium salt hydrate (BPS, Sigma-Aldrich, #146617-1G) for iron depletion (BPS-food) or 1 mM ferric ammonium citrate (FAC, Sigma-Aldrich, #F5879-100G) for iron enrichment (FAC-food). FAC-supplemented food was stored in the dark, whereas BPS media were stored under ambient light.

### Tissue collection, RNA isolation and library generation for RNA sequencing
To prepare the RNA-Seq samples, *Drosophila* $w^{1118}$ flies were reared for six generations on 120 μM BPS-supplemented (iron-depleted) media. Larvae of the 6th generation were carefully staged at the L2/L3 molt and transferred to either 1 mM FAC (iron-rich) or iron-depleted media (120 μM BPS). Brain-ring gland complexes (BRGC), whole gut and whole larvae (WB) were isolated from larvae reared on either media and were then collected at 4, 8, 12, and 16 h after the L2/L3 molt. Whole larval samples were washed in 1X PBS and gut samples were dissected in 1X PBS and transferred to TRIzol™ reagent (Invitrogen, #15596026). 20–25 gut and whole-body samples were used to extract RNA following the manufacturer's directions[91]. For BRGC, we dissected 30 specimens in 1X PBS and isolated RNA with the RNeasy Mini kit (Qiagen, #74104). RNA concentrations were measured with the Qubit 2.0 (Thermo Fisher Scientific) and the integrity of all samples was verified by the Agilent 2100 Bioanalyzer. We prepared 48 individual libraries (two replicates per each sample) with the Ovation RNA-Seq Systems 1–16 for Model Organisms kit (NuGEN, #0350). The RNA input for cDNA synthesis of each sample was 40 ng, and all cDNAs were fragmented prior to library generation. Fragmentation was carried out with the Covaris S220-Series System sonicator using a duty cycle of 10%, intensity at 5, 200 cycles, 180 s, 6 °C and a sample volume of 120 μl. The cDNA fragments were then ligated with provided barcode adapters for Illumina sequencing. Libraries were amplified by PCR, and their quality was assessed via the Agilent 2100 Bioanalyzer using Agilent High Sensitivity DNA 1000 Kit (Agilent, #5067-4626).

### Quantitative real-time PCR (qPCR)
All samples were collected in triplicate, and RNA was extracted as described above. All primer sequences are shown in Supplementary Data 1. RNA samples (0.25–1 μg/reaction) were reverse transcribed using the "High-Capacity cDNA Reverse Transcription" kit (Applied Biosystems™ #4374967). Synthesized cDNAs were diluted 1:20 for 1 μg total RNA per each reaction, and SYBR green Luna® Universal qPCR master mix (NEB #M3003L) was used for qPCR analysis in a Quant-Studio 6 Flex instrument (Applied Biosystems). The ΔΔCT method was used to normalize samples based on *rp49* expression. Statistical differences in expression were tested by the Student's *t* test from three biological replicates, and error bars represent 95% confidence intervals.

### Prussian blue iron staining
To stain for ferric iron, whole gut samples from 40 to 44-h 3rd instar larvae of *NP3084-Gal4* > RNAi animals were dissected in 1X PBS and immersed in fixation buffer (1X PBST with 4% Formaldehyde) for 25 min. Following three washes with 1X PBS for 10 min each, tissues were then incubated in 1% PBST for 20 min. Samples were then treated with Prussian blue stain for 45 min in 1% $K_4Fe(CN)_6$, 1% HCl in the dark at room temperature. Samples washed three times with PBS and were mounted in anti-fading mounting medium (Abcam, #AB188804) and photographed with a LEICA DFC500 camera.

### In situ hybridization
In situ hybridization probes were based on cDNAs obtained from the DGRC, and included *Mco4* (DGRC, #RE57944), *fire* (DGRC, #IP07844), *fire-like* (DGRC, #RH01238), *firewood* (DGRC, #IP06242), and *Mvl* (DGRC, #LD24465). The T7 promoter (5′- TAATACGACTCACTA-TAGGG −3′) was added to suitable regions of the ORF using Q5 DNA polymerase PCR amplification. Amplified fragments were purified via HighPrep PCR beads (MAGBIO, #AC-60005) and used as templates for probe synthesis by the SP6/T7 DIG RNA Labeling Kit (Sigma, #11175025910) following the manufacturer's instructions. Probes were purified via lithium chloride precipitation. Probe concentrations were determined with the Qubit™ RNA assay (Invitrogen, #Q32852), and probe quality was assessed via agarose gel electrophoresis. Gut samples from 40-h L3 were dissected in 1X PBS and fixed in 4% paraformaldehyde. The in situ hybridization procedure was carried out as described previously[92]. All solutions and buffer were created by RNase free water (Sigma, #4502-1L) and experiments

were carried out in a RNase free environment. In situ hybridization primers are presented in Supplementary Data 1.

## Larval gut pH assay

Acetazolamide (Sigma, #A6011-10g, final concentrations of 100 µM and 250 µM) and bromocresol purple sodium salt (Sigma, # 860891-5 g, final concentration 0.1% w/v) were added to Nutrifly medium prior to solidification[72]. 36-h L3 were fed for 4 h on media containing both the pH indicator and Acetazolamide, after which guts were dissected in 1X PBS buffer. Intestines were transferred to anti-fade mounting medium (Abcam, #AB188804), and all images were taken immediately with a Zeiss A1 brightfield microscope.

## X-ray fluorescence microscopy (XRF) synchrotron iron analysis

BRGCs were dissected from 40-h $w^{1118}$ L3 larvae in 0.25 M sucrose. Dissected BRGCs were transferred onto Thermanox cover slips and air-dried (Thermo Fisher Scientific # 50949476). X-ray fluorescence microscopy analysis (XRF) was performed at the Stanford Synchrotron Radiation light source (SSRL) at the SLAC National Accelerator Laboratory (https://www6.slac.stanford.edu), to generate elemental iron maps. All images were analyzed by the SMAK Microprobe Analysis Toolkit[93].

## Construction of CRISPR/Cas9 and transgenic lines

The *Mco4* knock-in (*3x-FLAG-Mco4*) and *Mco4* knockout (*Mco4$^{KO}$*) lines were generated by CRISPR/Cas9 homology-directed repair to produce FLAG-tagged Mco4 under endogenous control and a null deletion of the gene, respectively. Suitable genomic regions were obtained from FlyBase to design gRNAs via the CRISPR Optimal Target finder (http://targetfinder.flycrispr.neuro.brown.edu). All gRNA target sites were sequenced-verified by Sanger sequencing for the corresponding genomic regions in the Vas.Cas9 injection line (Bloomington #51323). Fragments corresponding to gRNAs were cloned into the pCFD5 plasmid (Addgene, #73914). All donor template fragments were amplified from genomic DNA of the Vas.Cas9 line via PCR and cloned into the backbone of the pDsRed-attP (Addgene, #51019) vector. For the knock-in line, the endogenous *Mco4* allele was replaced with a version carrying a C-terminal 3x-FLAG tag inserted immediately upstream of the stop codon in exon 4. For the knockout line, the Mco4 coding region was replaced with a 3xP3-DsRed marker cassette.

To generate the *UAS-Mco4-3Myc* transgenic line, a full-length *Mco4* cDNA (DGRC, #RE57944) was obtained from the *Drosophila* Genomic Resource Center. The *Mco4* wild-type cDNA was cloned into the PhiC31 pBID-UASC-GRM plasmid (Addgene, #35203). For the *fire, fire-like* double knockout, a transgenic gRNA line (*fire, fire-like*-gRNA) was generated based on four gRNAs (two gRNAs per gene) via the CRISPR Optimal Target finder (http://targetfinder.flycrispr.neuro.brown.edu) and cloning of the gRNAs into the pCFD5 plasmid (Addgene, #73914). pCFD5 is a PhiC31 vector with *vermilion* as a selectable marker, which expresses gRNAs under control of the U6-3 RNA Pol III snRNA promoter. To simplify screening for insertions, we replaced *vermilion* with *mini-white*. The double-mutant *fire$^{2xKO}$* line was generated by crossing UAS-fire, fire-like-gRNA with Act-Cas9 (Bloomington #54590). Mutant alleles were sequence-verified. Final plasmids and lines were confirmed by sequencing.

The *fire-mCherry, fire-like-eGFP* double knock-in line was generated using CRISPR/Cas9-mediated homology-directed repair. Two gRNAs – one upstream of *fire* and one downstream of *fire-like* – were designed based on results obtained via the CRISPR Optimal Target Finder. Target sites were sequence-verified via Sanger sequencing of the corresponding genomic loci in the Nos.Cas9 line (Bloomington #54591) and cloned into the pCFD5 plasmid (Addgene, #73914). Donor template fragments were PCR-amplified from the genomic DNA of the Nos.Cas9 line and cloned into the pDsRed-attP vector (Addgene, #51019), aimed at replacing the endogenous alleles of *fire* and *fire-like* genes with N-terminally tagged mCherry and eGFP, respectively.

A similar CRISPR/Cas9 strategy was used to generate the *firewood$^{KO}$* line. Here, the endogenous *firewood* locus was replaced with a DsRed marker flanked by two loxP sites. Two gRNAs targeting the upstream and downstream regions of *firewood* were designed with the CRISPR Optimal Target Finder tool and sequence-verified in the Nos.Cas9 line (Bloomington #54591). These were cloned into the pCFD5 plasmid (Addgene, #73914). Donor template fragments were PCR-amplified from Nos.Cas9 genomic DNA and inserted into the pDsRed-attP vector (Addgene, #51019). All plasmid fragments were amplified using Q5 High-Fidelity DNA Polymerase (NEB, #M0491S) and joined with the Gibson Assembly Master Mix (NEB, #E2611). All cloning and sequencing primers are provided in Supplementary Data 1.

## Yeast complementation assay

We used yeast strains BY4741 (wild-type) and Y16192 (ΔFet3). Cells were cultured in YPD broth (Sigma, #Y1375), YPDA plates (Sigma, #Y1500), and Synthetic Drop Out medium without uracil (SD-URA, Sigma, #Y1501-20g). Wild-type *Fet3* and *FTR1* genes were amplified from genomic BY4741 DNA and a full-length *Mco4* cDNA was obtained from the DGRC (#RE57944). Signal peptide sequences were identified with SignalP 5.091. We replaced the Mco4 secretion peptide with the first 40 amino acids of the Fet3p protein (Mco4$^{WT*}$). The Mco4 Glu267 to Ala (Mco4$^{E267A}$) mutant cDNA was derived from the wild-type Mco4$^{WT*}$ cDNA via PCR mutagenesis. The pESC-URA plasmid was used to construct pESC-URA-Mco4$^{WT*}$-FTR1, pESC-URA-*Mco4$^{E267A}$-FTR1*, and pESC-URA-*Fet3$^{WT}$-FTR1*. We then introduced these constructs into the ΔFet3 strain (Y16192) along with pESC-URA (negative control, empty vector) and the wild-type strain with pESC-URA (positive control, empty vector) using lithium acetate-mediated transformation[94], selecting for transformants on the appropriate synthetic drop-out medium. The transformed strains were verified via PCR amplification screening.

The complementation assay was carried out in SD-URA (no supplement) and in SD-URA with 80 µM or 160 µM BPS (iron-depleted medium). Overnight cultures from SD-URA (no supplement) were diluted to $OD_{600} = 0.2$ with sterile distilled water and incubated at 30 °C for ~14 h in appropriate media. ODs were measured in a Genesys™ 150 Spectrometer (Thermo Scientific). For the plate assay, 10 µL of overnight culture grown in SD-URA (no supplement, $OD_{600} = 0.2$) were inoculated on SD-URA and SD-URA/80 µM BPS plates. Growth phenotypes were evaluated after three days at 30 °C. All media contained 2% galactose. All yeast strains and plasmids were generously provided by Dr. David Stuart's lab at the University of Alberta, Edmonton, Canada. Fragments used for cloning were amplified using Q5 High-Fidelity DNA Polymerase (NEB, #M0491S), and plasmids were constructed using the Gibson Assembly Master Mix (NEB, #E2611). Yeast cell PCR screening was performed using Taq-DNA polymerase (NEB, #B9004S). All cloning and sequencing primers are listed in Supplementary Data 1.

## Inductively coupled plasma mass spectrometry (ICP-MS) analysis

Control ($w^{1118}$) and *Mco4* knockout mutant (*Mco4$^{KO}$*) animals were raised on a normal (non-supplemented) diet as well as a diet supplemented with 120 µM BPS (iron-depleted). Next, twenty 40-h L3 larvae were rinsed with MilliQ water. To purge ingested food from the gut, animals were kept in 1X PBS buffer for 30 min on a shaker with gentle agitation. Larvae were washed 3 × 15 min in MilliQ water to purge PBS buffer from the gut. Next, samples were digested in 1 mL of metal-free Nitric Acid (Thermo Fisher Scientific, #A509P212) at 200 °C using the CEM MARS6 microwave digestion system. ICP-MS analysis was performed using an Agilent 7900 ICP-MS instrument. Sample digestion and ICP-MS analysis

were conducted at The University of Alberta Biogeochemical Analytical Service Laboratory (BASL).

## Inductively coupled plasma optical emission spectrometry (ICP-OES)

Control ($w^{1118}$) and *fire*$^{2xKO}$ mutant flies were reared on two diets: standard Nutri-Fly food (non-supplemented) and Nutri-Fly food supplemented with 250 µM BPS (causes depletion of ferrous iron). For *Mvl*$^{97f}$ animals, both control ($w^{1118}$) and mutant flies were reared on a standard diet (12.5% w/v molasses, 10% w/v brewer's yeast, 1.6% w/v agar, 0.3% w/v gelatin, 1% w/v propionic acid, and 74.6% w/v water), with or without 250 µM BPS. Approximately 100 larvae (20-h L2 stage) were rinsed with MilliQ water and incubated in 1X PBS for 30 min with gentle agitation to purge ingested food. This was followed by three additional wash steps in MilliQ water to remove residual PBS. Samples were then digested in 1 mL of metal-free nitric acid (Thermo Fisher Scientific, #A509P212) at 200 °C using the CEM MARS6 microwave digestion system (CEM Corporation). Elemental analysis was performed using a PerkinElmer Optima 8300 ICP-OES instrument. Sample digestion and analysis were conducted at the Cinvestav in Mexico City, Mexico.

## Protein ferroxidase activity assay

The ferroxidase assay was based on the ceruloplasmin activity kit (Abcam, #273296) to assess the ferroxidase activity of Mco4 and HEPH proteins in *Drosophila* S2 cells. Full-length cDNA of *Drosophila Mco4* (DGRC, #RE57944) and human *HEPH* (GenScript, #OHu12228) were cloned into the pAFW plasmid with a C-terminal 3x Flag tag sequence. S2 cells were transfected using the Calcium-Phosphate transfection kit (Thermo Fisher Scientific, #K278001), and protein concentrations were assessed on Western blots. Transfected cells were harvested and washed 5x with 1X PBS buffer. Total proteins were then extracted using non-denaturing NP40 lysis buffer (Thermo Fisher Scientific, #J60766.AK), and protein concentrations were measured using the QubitTM Protein assay (Invitrogen, #Q33212).

We used 10 µL of the protein samples as input for the ceruloplasmin activity kit and followed the manufacturer's instructions. Samples were added to a 96-well flat-bottom UV-star plate (Sigma-Aldrich, # M3812-40EA), and the absorbance at 560 nm was measured via the VICTOR Nivo Multimode Microplate Reader. Parameters for standard and sample curves were determined using the formula:

Sample Activity = $S_k/S_s/V \times 2$ (mU/mL), where $S_k$ is the absorbance at the end and beginning of the linear range ($\Delta OD_{560}$) divided by the time $\Delta T$ representing end and beginning of the linear range in minutes). Ss denotes the slope of the standard curve (OD/nmol). V is the volume of the sample in ml, and 2 accounts for the dilution factor of the ammonium sulfate-precipitated samples.

## Immunostaining of gut samples

For Mco4 immunodetection we dissected gut samples in 1X PBS buffer from 40 to 44 h L3 *Mco4-3xFLAG* knock-in animals. Guts were fixed for 30 min in fixation buffer (1X PBST with 4% Formaldehyde) and washed 3 × 15 min with PBST (1X PBS with 0.1% Triton, Sigma, #T9284). Tissues were then blocked for 1 h in blocking solution (1X PBST with 5% Goat serum) and washed 3 × 15 min in PBST. Samples were incubated with the primary antibody solution (Rabbit anti-Flag from Cell signaling #14793S with 1:800 dilution in 1X PBST with 1% BSA) overnight at 4 °C on a rocking shaker. Samples were then rinsed 3× in 1X PBST followed by incubation with the secondary antibody for 1 h (Alexa Fluor 488, Abcam #ab150077 with 1:2000 dilution). Samples were rinsed 3× in 1X PBST and mounted in VECTASHIELD solution (Cell signaling #4083). All images were taken with a Nikon C2si Confocal Microscope.

## RNA-Seq data analysis

Sequencing services and raw FastQ data were provided by the Genome Quebec Innovation Center (http://www.genomequebec.com/en/innovation-center/). The quality of the FastQ files was evaluated with FastQC (version: v0.11.8)[95]. We then used HISAT2 to map the reads to the *Drosophila* reference genome (BDGP5 version)[96]. Next, SAMtool and HTSeq analyses were performed to create SAM files and quantifying read counts per gene, respectively[97,98]. The raw reads were then used for statistical analysis and to identify DEGs. We used Arraystar (version 7) and two well-known parametric R packages, Deseq2 and edgeR, to identify DEGs based on a rate-adjusted *p*-value < 0.05 for gene selection. We used MS Access to filter for significance, fold changes and cohort overlaps. Finally, we used R version 3.4.4 to run the Deseq2 and edgeR. All DEGs are presented in Supplementary Data 2. For term enrichment analysis, we used the Chi Square test to compare number of observed vs. expected genes. To identify genes linked iron or metal biology we assembled a reference list comprising 839 iron/metal genes (Supplementary Data 3). This list was compiled by filtering the fly transcriptome for all relevant gene function descriptors that referred to "iron" and other metal-related terms (for exact parameters see Supplementary Data 3), but we excluded the very large group of zinc finger transcription factors. Heatmaps were generated via ComplexHeatmap[99] and circlize[100] R packages. For DEGs clustering, we employed the K-means clustering algorithm with 1000 iterations and 100 runs to identify distinct expression profiles (clusters) in DEGs of BRGC, gut, and whole larval body samples. All clusters are presented in Supplementary Data 4.

## Mass spectrometry of transfected S2 cells

Mass spectrometry assay was carried out in *Drosophila* S2 cells as previously described[41]. All primers are presented in Supplementary Data 1. We used Schneider insect medium (Sigma, #RNBH8523) supplemented with 10% heat-inactivated fetal bovine serum and 1% streptomycin-penicillin to grow the S2 cells. Cells were cultured in 1 mM FAC or 120 µM BPS media. Plasmids carrying Hsp22 (DGRC, #LD36162) and Hsp70 (DGRC, #LP05203) cDNAs were obtained from the DGRC. Hsp22, Hsp70 and GFP ORFs were cloned into pAFW plasmid with a C-terminus 3× Flag tag sequence. All fragments were amplified via the Q5 High-Fidelity DNA Polymerase (NEB, #M0491S). Plasmids were constructed using the Gibson Assembly Master Mix (NEB, #E2611) and verified by sequencing. S2 cells were transfected with pAFW-Hsp22-3xFLAG, pAFW-Hsp70-3xFLAG, and pAFW-GFP-3xFLAG plasmids via the Calcium-Phosphate method (Thermo Fisher Scientific, #K278001). Cells were harvested after 48 h and washed three times with 1X PBS and lysed with lysis buffer (50 mM Tris-HCL, pH 7.4, 150 mM NaCl, 1 mM EDTA, 0.1% NP-40 and 1× proteinase K inhibitor). Proteins concentrations were measured via the QubitTM Protein assay (Invitrogen, #Q33212). To pull down proteins, we used 40 µl of M2 Flag beads (Sigma, #A2220). Beads were added to Chromotek columns (sct-50 spin) and incubated with cell lysates for 4 h at 4 °C on a rocking shaker following instructions of the manufacturer. Beads were then washed 2X with wash buffer #1 (25 mM Na-HEPES pH 7.5, 75 mM NaCl, 0.5 mM EDTA, 10% glycerol, 0.1% Triton X-100) followed by three washes with wash buffer #2 (25 mM Na-HEPES, pH 7.5, 75 mM NaCl, 0.5 mM EDTA, 10% glycerol). Pulled-down proteins were eluted by adding elution buffer to the beads, and incubation of columns at 95 °C for 5 min. Samples were separated on a 12% SDS gel and stained with Coomassie Brilliant Blue using standard protocol. SDS gel slices (containing pulled-down proteins) were cut for MALDI-TOF mass spectrometry (Alberta Proteomics and MS facility at the University of Alberta. The MS results are presented in Supplementary Data 5.

## Cloning, co-immunoprecipitation assay and Western blotting

Plasmids carrying cDNAs of Hsp22 (DGRC, #LD36162), Hsp70 (DGRC, #LP05203), mAcon1 (DGRC, #LD24561), and RFeSP (DGRC, #SD14047)

were obtained from the *Drosophila* Genomic Resource Center (DGRC; https://dgrc.bio.indiana.edu/Home). The Hsp22 and Hsp70 cDNA were cloned into pAFW plasmid so that they were in frame with the C-terminal sequence encoding the 3xFLAG. The pAFW-Hsp22-3xFLAG and pAFW-Hsp70-3xFLAG were then used to transfect S2 cells followed by immunoprecipitation. For co-immunoprecipitation and protein-protein interaction assays, the C terminus of mAcon1, RFeSP, and Fer1HCH DNA fragments were each fused to 5× Myc-tags. The tagged cDNAs were then cloned upstream of Hsp22-3xFLAG into the pAc5-STABLE-neo vector, which allows for simultaneously expressing two cDNAs: Both tagged cDNAs in a single plasmid are separated with T2A viral peptide, and upon cell transfection, the T2A is cleaved and allows equal production of either protein[101].

All cDNA fragments and plasmid backbones were PCR-amplified with Q5 High-Fidelity DNA Polymerase (New England Biolabs Inc. #M0491S). Amplified fragments were then fused to each other via the Gibson assembly master mix (New England Biolabs Inc. #E2611) according to the manufacturer's instructions. For co-immunoprecipitation and protein-protein interaction studies, the T2A sequence was added to the N-terminus of Hsp22-3FLAG and cloned into pAc5-STABLE-neo (the EGFP and NeoR/KanR fragments were removed). Plasmids were digested with KpnI (Thermo Fisher Scientific, #FD0524) and EcoRV (Thermo Fisher Scientific, # FD0303) in order to generate a backbone for the Gibson reaction with the mAcon1-5xMyc, RFeSP-5xMyc, and Fer1HCH-5xMyc fragments. Plasmids were cloned into competent DH5α *E. coli* cells. Prior to S2 cell transfection, all final plasmid constructs were validated via Sanger sequencing. All cloning and sequencing primers are showed in Supplementary Data 1.

*Drosophila* S2 cells were transfected with pAc5-STABLE-neo plasmids encoding Hsp22-3xFLAG (bait) and either Fer1HCH-5xMyc, mAcon1-5xMyc, or RFeSP-5xMyc (prey). After transfection, cells were washed and lysed, and lysates were incubated with M2 anti-FLAG agarose beads. Beads were washed twice with wash buffer #1 (25 mM Na-HEPES, pH 7.5; 75 mM NaCl; 0.5 mM EDTA; 10% glycerol; 0.1% Triton X-100), followed by three washes with wash buffer #2 (identical to buffer #1 but lacking Triton X-100). Bound protein complexes were then eluted for downstream analysis. The co-immunoprecipitated proteins were analyzed via Western blotting on a 12% SDS gel. To detect Flag-tagged and Myc-tagged proteins, we used monoclonal mouse anti-FLAG-tag (Cell Signaling, #8146S, 1:2000) and monoclonal rabbit anti-Myc-tag antibodies (Cell Signaling #2278S, 1:2000), respectively. Next, blots were incubated with goat anti-mouse IgG H&L HRP (Abcam #97023) and goat anti-rabbit IgG H&L HRP secondary antibodies (Abcam, #ab97051, 1:10,000). All blots were developed with the Amersham ECL Prime Western Blotting Detection reagent (Sigma, #GERPN2232) followed by scanning with the Bio-Rad Chemidoc™ MP imaging system.

### Ex vivo ferric reductase assay in *Drosophila* S2 cells
*Drosophila* S2 cells were used to assess ferric reductase activity in an ex vivo assay. Cells were cultured in Schneider's insect medium (Sigma, #RNBH8523) supplemented with 10% heat-inactivated fetal bovine serum and 1% Streptomycin-penicillin. Full-length *fire* (DGRC, # IP07844), *fire-like* (DGRC, # RH01238) and *firewood* (DGRC, #IP06242) cDNAs were cloned into the pAFW plasmid with a C-terminal 3xFLAG. A mutant *firewood* variant (designated *firewood**), in which the conserved heme-binding residues His39 and His63 were mutated to alanine, was generated by site-directed mutagenesis of the wild-type *firewood* cDNA. All plasmid fragments were PCR-amplified with Q5 High-Fidelity DNA Polymerase (NEB, #M0491S) and assembled via the Gibson assembly master mix (NEB, #E2611). All primer sequences are listed in Supplementary Data 1.

For the ferrozine assay, 5 ml cultures of S2 cells were transfected with pAFW-*fire*-3xFLAG, pAFW-*fire-like*-3xFLAG, pAFW-*firewood*-3xFLAG

or empty pAFW vector using Lipofectamine™ 3000 Reagent (Thermo Fisher Scientific, #L3000001), according to the manufacturer's instructions. After 72 h, cells were harvested and washed three times in 1X PBS. For co-transfection experiments, plasmids were mixed in a 1:1 ratio prior to transfection.

Proteins were extracted using NP-40 lysis buffer (Thermo Fisher Scientific, #J60766.AP) supplemented with protease inhibitors (Sigma, # 11836170001). Cells were vortexed for 30 s, and briefly centrifuged ($2350 \times g$), for a total of five times. Cell debris was not removed. Protein concentrations were determined using the Qubit™ Protein assay kit (Invitrogen, #Q33212). For each reaction, 50 μL of cell lysate, 50 μL of freshly prepared 1 mM ferric ammonium citrate (Sigma #F5879), and 50 μL of 10 mM ferrozine solution were combined in a 96-well flat-bottom microplate (Sigma-Aldrich, # M3812-40EA). Samples were incubated for 1 h at room temperature on a shaker, and absorbance at 562 nm was measured using a VICTOR Nivo Multimode Microplate Reader.

To generate a standard curve, ferrous standards (ammonium iron (II) sulfate, Sigma, #09719-50 G) were prepared at final concentrations of 0, 10, 20, 30, 40, and 50 μM. Each standard (50 μL) was mixed with 50 μL of 10 mM ferrozine solution, and absorbance was measured at 562 nm. Sample absorbance values were converted to $Fe^{2+}$ concentrations using the linear equation from the standard curve ($y = mx + c$).

### Reporting summary
Further information on research design is available in the Nature Portfolio Reporting Summary linked to this article.

### Data availability
The source data underlying all figures are provided. RNA sequencing raw data related to Fig. 1D, Fig. 2, Supplementary Fig. S2, Table 1, Table 2, Supplementary Data 2, and Supplementary Data 4 have been deposited in the NCBI Sequence Read Archive (SRA) under accession number PRJNA1149758. Mass spectrometry proteomics raw data related to Supplementary Fig. S3, Supplementary Data 5, and Supplementary Tables 3 and 4 have been deposited with the ProteomeXchange Consortium via the PRIDE partner repository (under accession #PXD055043). All other data supporting the findings of this study are available from the corresponding author upon request. Source data are provided with this paper.

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

## Acknowledgements

The authors thank Michael O'Connor for providing fly stocks. We would also like to extend our gratitude to Dr. David Stuart for his guidance and providing yeast strains, plasmids and reagents. We also wish to thank Dr. Kacie Norton for allowing us to use the equipment and services at the Advanced Microscopy Facility (AMF) at the University of Alberta. We also wish to thank the Bloomington *Drosophila* Stock Center at Indiana University, Kyoto *Drosophila* Stock Center and the Vienna *Drosophila* Resource Center for sending fly stocks. Synchrotron analysis via the use of the Stanford Synchrotron Radiation Lightsource and the Linac Coherent Light Source, SLAC National Accelerator Laboratory, is supported by the US Department of Energy, Office of Science, Office of Basic Energy Sciences under contract No. DE-AC02-76SF00515. The SSRL Structural Molecular Biology Program is supported by the DOE Office of Biological and Environmental Research, and by the National Institutes of Health, National Institute of General Medical Sciences (Grant P30GM133894). K.K.J. wishes to thank the CIHR (PS 169102) and NSERC (RGPIN-2018-04357) for supporting this work.

## Author contributions

S.S. co-designed the experiments, conducted most of the experiments, and wrote the initial manuscript draft. M.Y. performed fly screening for *Fire* $^{2xKO}$ mutant animals and cloning for the double *Fire-mCherry, Fire-like-EGFP* knock-in line, *Firewood* $^{KO}$ line, and the yeast *Mco4* $^{E267A}$ plasmid. Q.U. generated the *Mco4-3XFLAG* knock-in line and conducted the ferroxidase assay and the survival analysis of control flies in BPS- and FAC-supplemented diets. A.A.E. generated the *Mco4* $^{KO}$ and *UAS-Mco4-myc* lines. E.P. performed cloning for the double *Fire-mCherry, Fire-like-EGFP* knock-in line. S.L.W. and T.K. conducted Synchrotron iron analysis at the Stanford Synchrotron Radiation Lightsource. Y.M.B.L. performed ICP-OES experiment of *Fire* $^{2xKO}$ mutant animals, larval collection, and ICP-OES analysis of *Mvl97f* mutant animals. F.M. co-designed the experiments and co-wrote the manuscript with input from the other authors. K.K.J. supervised and co-designed the experiments and co-wrote the manuscript with input from the other authors.

## Competing interests

The authors declare no competing interests.
