## [Peer Review file · Nature Communications]

New molecular components of high and low affinity iron import systems in *Drosophila*

Corresponding Author: Professor Kirst King-Jones

Version 0:

Reviewer comments:

Reviewer #1

(Remarks to the Author)

The manuscript by Soltani reports very significant novel findings concerning iron metabolism in *Drosophila*. It is very well written and conveys the principal findings of this complex study clearly. The authors used a very elegant and interesting strategy of gentle iron starvation followed by sudden exposure to iron, enabling them to detect genes involved in iron uptake (down-regulated) and genes involved in iron transport, storage and detoxification (up-regulated). This led to the identification of four new genes involved in iron absorption, including, most strikingly, a high affinity iron uptake system involving *Mco4*, a *Fet3* orthologue, the first instance of this in animals. Impressive!

I should say that I am not an expert in the transcriptomics or microscopy methods employed in this study, but my impression is that the experiments were carefully conducted with suitable controls. Impressively, the authors used multiple lines of evidence from different experiments to arrive at their conclusions.

I think this is a very important manuscript that should be published. I have only minor comments.

Line 79 Perhaps I missed it but I think prothoracic gland is abbreviated to PG in the manuscript but I could not find this defined anywhere.

Line 538 Please don't perpetuate the myth that ferritins store 4500 irons. This was an estimate, based on many assumptions that may or may not be correct, of capacity. No ferritin has been observed containing this much iron. Better to say that ferritins can store large amounts of iron (hundreds to thousands of ferric ions).

Line 579 Strictly speaking, this should be 'electron' singular.

Figure 2 x-axis needs units (hrs?). Same for Fig. S3, S4.

Line 607 Why would genes encoding metallothioneins be amongst the early responders in whole body samples? Presumably this is not associated with a detoxification function?

Line 792 '.....on 120 μ m BPS media....' Edit to capital M for micromolar.

Line 882 This is indeed the case. When we reared control or *tub>Mco4-cDNA* animals on 500 μ M BPS, controls were unable to survive larval development, and never formed pupae. By contrast, *tub>Mco4-cDNA* animals not only formed pupae (Fig. 6E), but 40% of the population reached adulthood (Fig. 6F).

I have perhaps misunderstood something here, but *tub>Mco4-cDNA* animals on 500 micromolar BPS did not survive but then in the second part, 40% reached adulthood. Both are with iron chelator. What is the difference here? Please clarify.

Reviewer #2

(Remarks to the Author)

The authors describe a novel screen for genes regulated by iron deficiency in the fly larva intestine and brain ring gland complex (BRGC) and identify two cytochrome b561 homologues, a cytochrome b5 homologue, and a protein homologous to the yeast ferroxidase *Fet3P* that are upregulated in multigenerational iron deficiency and necessary to support optimal

growth/survival in low iron conditions.

The methods are exceptionally well-described and the findings are similarly well narrated and illustrated and often corroborated by orthogonal approaches.

My concerns with the manuscript really only lie in the significance of the findings and whether the data presented actually support the conclusions, particularly that the ferroxidase (MCO4) is part of a high affinity iron uptake “complex.”

Each of the proteins that is a focus of this manuscript is homologous to a protein known to be involved in iron metabolism in other species, which detracts from the overall significance of the paper—in fact, the screen, while elegant, really wasn’t necessary to get to these proteins as candidates. The screen allowed them to single out these proteins, among several other orthologues in the fly genome, as more likely to be relevant to iron metabolism, which they validate with functional studies.

The Cyb561 homologues (fire and fire-like) are similar to DCYTB, thought to be involved in iron uptake in the mammalian gut by reducing the abundant form of iron, Fe³⁺. The authors suggest (perhaps too strongly) that the Cyb5 homologue is the molecule that provides the electrons to the Cyb5 homologues to reduce iron. Though they provide some in vivo expression/colocalization/coregulation and functional evidence in RNAi/deletional strains suggesting that they together function together, an ex vivo functional study demonstrating similar co-dependence would go further to indicate that this is direct and not through some more complicated functional interaction. Using a mutant Cytb (firewood) protein, unable to generate electrons would go further to show that the hypothesized function is indeed true. Similarly, coimmunoprecipitation would further cement their codependence.

In yeast, the MCO4 orthologue, Fet3p, pairs with Ftr1, a transmembrane protein, to transport Fe³⁺ across the membrane. To call MCO4 part of a similar “system” or “complex”. The authors really need to show what the transmembrane transporter is. Is it a ZIP family transporter? If so, which one?

Secondarily, because the actual mechanism by when ABCE and Evi5, particularly the former, disrupt iron metabolism is unknown, the authors should be careful in interpreting iron-dependence of the upregulation of MCO4 in the context of deletion of either of these two genes.

Reviewer #3

(Remarks to the Author)

In this manuscript, Soltani et al. identified a few new components of iron uptake systems in the model animal *Drosophila*. The authors first conducted an RNA-seq experiment on fly larvae as well as the gut and brain ring gland complex from the larvae that had been subjected to long-term iron depletion followed by short-term iron repletion. The authors then proceeded to study the function of several candidate genes based on their transcriptional responses to iron levels and their homology to known proteins involved in iron transport. Three previously uncharacterized genes, Fire, Fire-like, and Firewood, were shown to be part of a low-affinity iron uptake system. They are likely to serve as ferric reductases and the associated electron donor for reducing ferric iron for the subsequent transport by the divalent metal transporter Mvl in the *Drosophila* gut. The fourth protein Mco4 was found to be a ferroxidase important for the high-affinity iron uptake; it was demonstrated to have the same function as the yeast homolog Fet3p.

The molecular biology and genetics experiments in this manuscript are solid and the proposed role of Mco4 is well supported by appropriate technical approaches. With regard to the candidate genes in the low-affinity iron transport systems, additional cell biological and biochemical experiments may help to support the conclusion in the manuscript.

Major comments:

1 Where do these proteins localize in cells? Given their proposed function in iron uptake, these proteins (with the exception of Firewood) are presumed to present on the plasma membrane. This point needs to be verified by experiments.

2 The authors proposed that: “Fire and Fire-like are the principal gut ferric reductases in *Drosophila*”. In Figure 4C, they showed that deficiency of these two genes led to reduced formation of BPS-Fe²⁺ precipitates in the gut. This result is a good indication that these proteins may be ferric reductases, however, the reduced BPS-Fe²⁺ precipitates could also be an indirect outcome of changes in pH or impairment of downstream iron transport system. Other biochemical experiments are needed to validate the ferric reductase activity of these proteins.

3 Besides the reported reduction of iron in the gut, are the organismal iron levels decreased in the larvae deficient of fire, fire-like, or firewood? If so, is the degree of decrease comparable to that of Mvl RNAi samples? These results may help to determine the contribution to these proteins to the iron homeostasis in *Drosophila* under a certain condition.

4 The physical interaction between Mvl and Fire, Fire-like, or Firewood would be a direct evidence to support that they function in the same complex/system. The same strategy used for Figure S3C in this manuscript can be used to examine the interaction among these proteins.

Minor comments:

1 Mco4 displayed the highest expression in proventriculus. It was even reported as one of the marker genes for a foregut cell type. Does Mco4 help to reduce iron in proventriculus, or does it play an iron-independent role?

2 In Figure 3, panel A shows Prussian blue staining results under two conditions but panel B shows only one set of data. Please indicate which condition is summarized in panel 3B.

3 Do fire and fire-like play slightly different roles? In Figure 4C, fire RNAi induced a phenotype but RNAi against fire-like did not. Do their single knockout lines have reduced staining of BPS-Fe²⁺ precipitates or other phenotypes?

4 There are some typos in the text.

e.g. line 363: "BPST"

line 1073: "ph".

Version 1:

Reviewer comments:

Reviewer #1

(Remarks to the Author)

The authors have addressed all of my comments in their response and the revised manuscript.

Reviewer #2

(Remarks to the Author)

The authors have done an excellent job responding to my comments and have integrated new data when appropriate.

Reviewer #3

(Remarks to the Author)

The authors have addressed our concerns and comments properly. One very minor point that has not been fixed is the typo "BPST" in line 394.

Responses to the Reviewers' comments

We thank the reviewers and the editor for carefully reading the manuscript and providing constructive feedback. We have revised the manuscript accordingly and believe that the changes have improved its clarity and strength. Major new text sections in the main manuscript are highlighted in red font and our responses to the reviewers are shown in blue in this document.

Reviewer #1 (Remarks to the Author):

The manuscript by Soltani reports very significant novel findings concerning iron metabolism in *Drosophila*. It is very well written and conveys the principal findings of this complex study clearly. The authors used a very elegant and interesting strategy of gentle iron starvation followed by sudden exposure to iron, enabling them to detect genes involved in iron uptake (down-regulated) and genes involved in iron transport, storage and detoxification (up-regulated). This led to the identification of four new genes involved in iron absorption, including, most strikingly, a high affinity iron uptake system involving Mco4, a Fet3 orthologue, the first instance of this in animals. Impressive!

I should say that I am not an expert in the transcriptomics or microscopy methods employed in this study, but my impression is that the experiments were carefully conducted with suitable controls. Impressively, the authors used multiple lines of evidence from different experiments to arrive at their conclusions.

I think this is a very important manuscript that should be published. I have only minor comments.

We sincerely thank the reviewer for the encouraging comments and thoughtful assessment. We are pleased that the study's importance and our multi-pronged experimental strategy were appreciated.

Line 79 Perhaps I missed it but I think prothoracic gland is abbreviated to PG in the manuscript but I could not find this defined anywhere.

Line 79 now reads: ...hormone-producing glands (such as the *Drosophila* prothoracic gland, hereafter "PG")

Line 538 Please don't perpetuate the myth that ferritins store 4500 irons. This was an estimate, based on many assumptions that may or may not be correct, of capacity. No ferritin has been observed containing this much iron. Better to say that ferritins can store large amounts of iron (hundreds to thousands of ferric ions).

That's a fair point! Sentence now states: "Ferritins are molecular nanocages that store excess cytosolic iron, with a capacity of storing hundreds to thousands of oxidized iron atoms per cage".

Line 579 Strictly speaking, this should be 'electron' singular.

Done.

Figure 2 x-axis needs units (hrs?). Same for Fig. S3, S4.

The figures have been updated accordingly.

Line 607 Why would genes encoding metallothioneins be amongst the early responders in whole body samples? Presumably this is not associated with a detoxification function?

Our data indicate spatially distinct regulation among the *Drosophila* metallothioneins: *MtnB* is strongly induced in the gut, whereas *MtnA*, *MtnD*, and *MtnE* are upregulated in whole-body samples but not in the gut cohort. A 2017 study (Wenja et al.) reported gut expression and iron inducibility of *MtnB*, supporting its limited role in iron detoxification. However, other metallothioneins may be responding indirectly. One possibility is that iron supplementation perturbs metal homeostasis, leading to cellular zinc accumulation, which in turn induces metallothionein expression. It is also possible that iron supplementation increases oxidative stress, and metallothioneins are known for their ability to interact with reactive oxygen species (ROS) and redox-active metals. Based on our data, the five *Drosophila* metallothionein genes appear to have tissue-specific functions, though the underlying reasons for this remain unclear.

Line 792 ‘.....on 120 μM BPS media....’ Edit to capital M for micromolar.

Revised.

Line 882 This is indeed the case. When we reared control or *tub>Mco4-cDNA* animals on 500 μM BPS, controls were unable to survive larval development, and never formed pupae. By contrast, *tub>Mco4-cDNA* animals not only formed pupae (Fig. 6E), but 40% of the population reached adulthood (Fig. 6F).

I have perhaps misunderstood something here, but *tub>Mco4-cDNA* animals on 500 micromolar BPS did not survive but then in the second part, 40% reached adulthood. Both are with iron chelator. What is the difference here? Please clarify.

Thank you for pointing this out — we agree that the original phrasing may have caused confusion. To clarify, both genotypes were indeed exposed to 500 μM BPS, but their responses were different because overexpression of *Mco4* successfully competes with BPS for iron! We have revised the sentence as follows:

“When we reared animals on food supplemented with 500 μM BPS, control animals failed to survive larval development and never formed pupae. Remarkably, *tub>Mco4-cDNA* animals were partially rescued under the same conditions, with successful pupation (Fig. 6E), and 40% reaching adulthood (Fig. 6F).”

Reviewer #2 (Remarks to the Author):

The authors describe a novel screen for genes regulated by iron deficiency in the fly larva intestine and brain ring gland complex (BRGC) and identify two cytochrome b561 homologues, a cytochrome b5 homologue, and a protein homologous to the yeast ferroxidase Fet3P that are upregulated in multigenerational iron deficiency and necessary to support optimal growth/survival in low iron conditions.

The methods are exceptionally well-described and the findings are similarly well narrated and illustrated and often corroborated by orthogonal approaches.

Thank you for your kind words!

My concerns with the manuscript really only lie in the significance of the findings and whether the data presented actually support the conclusions, particularly that the ferroxidase (MCO4) is part of a high affinity iron uptake “complex.”

We agree that additional data, particularly biochemical evidence, will be necessary to unequivocally establish that *Mco4* is part of a high-affinity iron uptake complex. At present, we have laid the genetic groundwork for future studies to explore this hypothesis. That said, the genetic data are compelling, in particular the finding that overexpression of *Mco4* enables larval survival on severely iron-depleted media that is otherwise lethal to wild type controls. This finding implies that elevated *Mco4* expression mitigates the effects of iron starvation, most likely by enhancing dietary iron uptake. Consistent with this, we observed increased ferritin accumulation in the gut of *Mco4*-overexpressing animals. In addition, we demonstrated that *Drosophila Mco4* can rescue yeast *Fet3* mutants, indicating conserved evolutionary function - despite a separation of > one billion years.

We would also like to emphasize that this study was conceived as a “Resource” article, providing the iron biology community with a comprehensive dataset of novel candidate genes involved in iron metabolism. (Although *Nature Communications* does not formally offer this article type, this remains the intended spirit of the work.) Rather than subdividing this work into multiple papers focused on individual genes, we chose a systems-level approach - selecting candidates with strong expression and interesting transcriptional profiles as proof of principle - to provide a solid foundation for future mechanistic studies.

Each of the proteins that is a focus of this manuscript is homologous to a protein known to be involved in iron metabolism in other species, which detracts from the overall significance of the paper—in fact, the screen, while elegant, really wasn’t necessary to get to these proteins as candidates. The screen allowed them to single out these proteins, among several other orthologues in the fly genome, as more likely to be relevant to iron metabolism, which they validate with functional studies.

As outlined above, the goal of this work was to provide a resource for the iron biology community, and contribute to both human and *Drosophila* model systems. Importantly, this project was initiated without

a specific focus on iron absorption - our early interest centred on *Mco4* and the *Hsp70* genes. The emphasis on ferric reductases only emerged several years later, based on where the data led us.

While it is true that alternative approaches could have led to the same candidates, that outcome was far from predetermined, and the strength of our approach lies in its unbiased design. Moreover, the RNA-Seq dataset yielded far more than just the genes we highlight here - to name a few examples: (i) dozens of additional candidates with promising iron-responsive profiles, (ii) co-regulation patterns (e.g., the *Fire* genes clustering with *Malvolio/DMT1*), (iii) tissue-specific and temporal dynamics of transcriptional responses, and iv) hierarchical context (e.g., identifying which genes show the strongest responses).

Regarding the cytochrome b561 homologues: these were not previously on the radar. In fact, prior reviews typically emphasized *CG1275* and *nemy* as the only plausible *DCYTB* homologs in *Drosophila*. The identification of *fire* and *fire-like* was unexpected and had not been proposed previously. Granted, a researcher with a focus on ferric reductases could have examined all eight ferric reductase genes manually, but nobody in the field has done this. The RNA-Seq strategy not only enabled us to detect these genes with high confidence but provided critical context regarding their regulation and function. Finally, we would likely never have investigated *firewood* (cytochrome b5) without this screen. Its identification as a co-regulated gene - and later, a candidate partner in ferric reduction - challenges prevailing models that assume ascorbate as the sole electron donor and has opened new avenues of inquiry.

The Cyb561 homologues (*fire* and *fire-like*) are similar to *DCYTB*, thought to be involved in iron uptake in the mammalian gut by reducing the abundant form of iron, Fe^{3+} . The authors suggest (perhaps too strongly) that the *Cyb5* homologue is the molecule that provides the electrons to the *Cyb5* homologues to reduce iron. Though they provide some *in vivo* expression/colocalization/coregulation and functional evidence in RNAi/deletional strains suggesting that they together function together, an *ex vivo* functional study demonstrating similar co-dependence would go further to indicate that this is direct and not through some more complicated functional interaction. Using a mutant *Cytb* (*firewood*) protein, unable to generate electrons would go further to show that the hypothesized function is indeed true. Similarly, coimmunoprecipitation would further cement their codependence.

Thank you for this important question! We have taken several steps to address this point.

(i) We conducted an *ex vivo* ferric reductase assay in S2 cells using ferrozine. To this end, we generated two constructs: one expressing a wild-type *firewood* (*firewood^{WT}*) cDNA and another expressing a mutant version (*firewood^{**}*), in which the conserved histidine residues (H39 and H63) were replaced with alanine to disrupt electron transfer. The two point mutations abolish heme binding and thus prevent electron donation.

We then measured the ferric reductase activity of *Fire* and *Fire-like* proteins in the presence or absence of either wild-type or mutant *Firewood*. Co-transfection of *fire/fire-like* with *firewood^{WT}* resulted in 50% higher ferric reduction compared to *fire/fire-like* alone or when co-transfected with the *firewood^{**}* mutant (Figure 4L, Figures S6C and S6D). These results strongly support the idea that *Fire*, *Fire-like*, and *Firewood* function together in ferric iron reduction.

ii) In addition, we generated a CRISPR knockout of firewood (*firewood^{KO}*) to eliminate the endogenous locus (Figures S6A and S6B). We examined BPS-Fe²⁺ precipitates in gut samples from *firewood^{KO}* animals and observed that, similar to *tub>firewood-RNAi* larvae, *firewood^{KO}* mutants failed to form precipitates (Figure 4C, Figure S7). This provides independent genetic evidence that Firewood is required for dietary iron reduction in the gut.

With respect to the co-immunoprecipitation experiment: We agree this is an excellent suggestion. However, we plan to address it in a future study. Specifically, a PhD student in the lab is currently generating a tagged knock-in line for Firewood, which will enable co-immunoprecipitation experiments using in vivo samples. In parallel, we are initiating TurboID-based proximity labeling for all three fire genes as part of a follow-up project.

In yeast, the MCO4 orthologue, Fet3p, pairs with Ftr1, a transmembrane protein, to transport Fe³⁺ across the membrane. To call MCO4 part of a similar “system” or “complex”. The authors really need to show what the transmembrane transporter is. Is it a ZIP family transporter? If so, which one?

We appreciate the reviewer’s point and would like to clarify our reasoning for using the term “complex” or “system”.

Mco4, like its yeast counterpart Fet3p, is a multicopper oxidase and, by itself, has no known capacity to import iron across membranes. If Mco4 is indeed involved in high-affinity iron uptake, as our genetic data suggest, it would necessarily require additional membrane-associated partners to function. By using the term “complex” (or “system”), we aimed to clearly convey to readers that Mco4 alone is unlikely to mediate iron transport and that additional components must be involved.

We acknowledge that we are inferring the existence of a larger protein complex based primarily on the strong evolutionary conservation between Mco4 and Fet3p. However, we agree that identifying the transmembrane transporter partner for Mco4 is a critical next step.

Regarding the possibility of a ZIP family transporter serving as the Ftr1 equivalent: this is an intriguing hypothesis, especially given that this study identified a few candidate ZIP genes. However, pinpointing the exact partner will require extensive additional work, including in vivo TurboID experiments using tagged Mco4, mutant studies of candidate transporter genes, and biochemical analyses to demonstrate iron transport, as well as physical interaction and functional cooperation with Mco4. It is our hope that we will be able to address this important question in a future study.

Secondarily, because the actual mechanism by which ABCE and Evi5, particularly the former, disrupt iron metabolism is unknown, the authors should be careful in interpreting iron-dependence of the upregulation of MCO4 in the context of deletion of either of these two genes.

Thank you for pointing this out! Regarding AGBE, we have previously published evidence that AGBE physically interacts with IRP1 and mitoNEET, contributing to the repair of oxidatively damaged Fe-S clusters that otherwise render IRP1 non-functional. While several mechanistic aspects of this pathway

remain to be elucidated, it is clear that loss of *AGBE* disrupts *IRP1* function.

Importantly, disruption of *AGBE*, *IRP1*, or *mitoNEET* leads to highly similar phenotypes, including iron depletion in the prothoracic gland and porphyria-like autofluorescence. Increased iron availability in the prothoracic gland reduces autofluorescence in these mutants, whereas iron depletion exacerbates it. Thus, despite gaps in the mechanistic details, autofluorescence in this context serves as a robust and quantifiable reporter of intracellular iron status (further supported by *Alas* transcript levels measured by qPCR).

To add further confidence to our interpretation, we also examined *Mco4* expression in *Evi5*-depleted ring glands, which exhibit iron starvation through an entirely different mechanism - namely, a block in intracellular iron transport. In this context as well, we observed strong *Mco4* induction, corroborating the findings from the *AGBE* loss-of-function background.

In interpreting the upregulation of *Mco4* in the context of *AGBE* or *Evi5* depletion, we have remained mindful of these mechanistic uncertainties and have framed our conclusions accordingly. Specifically, we have revised the text from: “These genetic data demonstrate that *Mco4* plays a role in cellular iron availability, strongly suggesting a role in cellular iron import into the PG.” with “These genetic data suggest that *Mco4* contributes to maintaining cellular iron availability in the PG and are consistent with a role in facilitating iron import” (lines 929-930). We hope that this rewording appropriately conveys the strength and limitations of the available data.

Reviewer #3 (Remarks to the Author):

In this manuscript, Soltani et al. identified a few new components of iron uptake systems in the model animal *Drosophila*. The authors first conducted an RNA-seq experiment on fly larvae as well as the gut and brain ring gland complex from the larvae that had been subjected to long-term iron depletion followed by short-term iron repletion. The authors then proceeded to study the function of several candidate genes based on their transcriptional responses to iron levels and their homology to known proteins involved in iron transport. Three previously uncharacterized genes, *Fire*, *Fire-like*, and *Firewood*, were shown to be part of a low-affinity iron uptake system. They are likely to serve as ferric reductases and the associated electron donor for reducing ferric iron for the subsequent transport by the divalent metal transporter *Mvl* in the *Drosophila* gut. The fourth protein *Mco4* was found to be a ferroxidase important for the high-affinity iron uptake; it was demonstrated to have the same function as the yeast homolog *Fet3p*.

The molecular biology and genetics experiments in this manuscript are solid and the proposed role of *Mco4* is well supported by appropriate technical approaches. With regard to the candidate genes in the low-affinity iron transport systems, additional cell biological and biochemical experiments may help to support the conclusion in the manuscript.

Thank you for your positive assessment. We have addressed the additional points regarding the low-affinity system with new experiments, as outlined below.

Major comments:

1 Where do these proteins localize in cells? Given their proposed function in iron uptake, these proteins (with the exception of Firewood) are presumed to present on the plasma membrane. This point needs to be verified by experiments.

Thank you for this great suggestion! We agree that subcellular localization is critical for understanding the role of ferric reductases in iron uptake. To address this, we generated three new CRISPR knock-in lines: (1) fire-mCherry, (2) fire-like-eGFP, and (3) a double knock-in line co-expressing both fluorescently tagged proteins. We assessed all three lines and present results for the double knock-in line, which allows simultaneous visualization of both proteins. The expression patterns of Fire and Fire-like mirrored the RNA in situ hybridization results (Fig. 4D), with both transcripts being detected upstream (anterior midgut) and downstream (iron region and posterior midgut) of the BPR region (Fig. 4E, Fig. S8B).

Under iron-depleted conditions, Fire protein localized to the apical membrane of anterior and posterior midgut cells, consistent with a role in luminal ferric iron reduction. In the iron region, strong *fire* expression obscured distinct membrane labeling in some cells, though membrane localization could still be observed in enhanced Z-projection images (Fig. 4E, Fig. S8D–E) (see image below, i.e., S8E).

Fire-like displayed a similar regional distribution but at much lower levels. It was primarily detected in the anterior midgut and weakly in the posterior midgut, but absent from the iron region (Fig. S8B). Due to its low expression, distinct membrane localization was not consistently observed. Instead, fire-like appeared diffusely distributed between cytosol and membrane (Fig. 4E), raising the possibility that it may function in endosomal or lysosomal compartments.

These expression differences of the knock-ins are consistent with the transcriptional upregulation under

BPS treatment of their wild type counterparts (*fire*: ~1,100-fold; *fire-like*: ~21-fold, Fig. 4M), indicating that *fire* is expressed at levels ~52-fold higher than *fire-like*.

2 The authors proposed that: "Fire and Fire-like are the principal gut ferric reductases in *Drosophila*". In Figure 4C, they showed that deficiency of these two genes led to reduced formation of BPS-Fe²⁺ precipitates in the gut. This result is a good indication that these proteins may be ferric reductases, however, the reduced BPS-Fe²⁺ precipitates could also be an indirect outcome of changes in pH or impairment of downstream iron transport system. Other biochemical experiments are needed to validate the ferric reductase activity of these proteins.

Agreed! To directly test the ferric reductase activity of *fire* and *fire-like*, we conducted an ex vivo assay in S2 cells using ferrozine. Expression of either *fire* or *fire-like* in S2 cells resulted in significantly increased ferric reduction compared to cells transfected with an empty control plasmid, supporting their function as ferric reductases.

We further examined whether this activity was enhanced by Firewood, their putative electron donor. Co-transfection of *fire* or *fire-like* with wild-type *firewood* (*firewood*^{WT}) led to ~50% higher ferric reductase activity, whereas no enhancement was observed with a mutant *firewood* construct (*firewood*^{**}), in which histidine residues required for electron transfer (H39 and H63) were replaced with alanine (Figure 4K, Figures S6C and S6D).

While all data presented in the manuscript were obtained from cell lysates, we also performed assays on intact S2 cells to assess whether Fire and Fire-like proteins are accessible on the outer plasma membrane. In these experiments (not shown in the manuscript), cells expressing *fire* or *fire-like* exhibited ~2-fold higher ferric reductase activity (see figure below) compared to control cells, consistent with surface localization and functional activity at the cell membrane.

Taken together, these findings confirm that Fire and Fire-like act as plasma membrane-associated ferric reductases and that Firewood enhances their activity.

3 Besides the reported reduction of iron in the gut, are the organismal iron levels decreased in the larvae deficient of *fire*, *fire-like*, or *firewood*? If so, is the degree of decrease comparable to that of *Mvl* RNAi samples? These results may help to determine the contribution to these proteins to the iron homeostasis in *Drosophila* under a certain condition.

We agree that comparing organismal iron levels across genotypes is important for understanding their contributions to systemic iron homeostasis. However, performing this analysis required overcoming some experimental limitations.

Both control and *fire*^{2x[^]KO} animals are viable on standard food and on food containing 120 μM BPS. To increase sensitivity, we opted for a more stringent iron-depleted condition (250 μM BPS). However, at this concentration, larval lethality was high for both genotypes (Figure 4I), making it difficult to obtain sufficient third instar larvae for analysis.

We therefore collected late second instar (L2) larvae, obtaining five biological replicates (~100–150 larvae each) for both control and *fire*^{2x[^]KO} animals reared on standard and 250 μM BPS diets.

As a point of reference, we also assessed organismal iron levels in *Mvl*^{97f} mutants under similar conditions. All samples were analyzed using Inductively Coupled Plasma Optical Emission Spectroscopy (ICP-OES), and the results are presented in Figure 4J.

These measurements revealed that *fire*^{2x[^]KO} animals exhibited a significant reduction in whole-body iron levels under both normal and iron-depleted conditions, comparable in magnitude to *Mvl*-deficient larvae on standard food. On 250 μM BPS, however, *Mvl* mutants did not show further reduction—consistent with the complete loss of transporter activity under both conditions.

These results support the conclusion that *fire* and *fire-like* are required to maintain normal systemic iron levels, consistent with their proposed role in dietary iron absorption.

4 The physical interaction between Mvl and Fire, Fire-like, or Firewood would be a direct evidence to support that they function in the same complex/system. The same strategy used for Figure S3C in this manuscript can be used to examine the interaction among these proteins.

We agree that detecting a physical interaction between Mvl and Fire, Fire-like, or Firewood would provide direct evidence for a functional complex. While this remains an intriguing possibility, it is worth noting that in vertebrates, a direct interaction between DCYTB and DMT1 has not been demonstrated.

Nevertheless, we tested for a physical interaction between Mvl and Fire or Fire-like using co-immunoprecipitation in S2 cells. These experiments did not reveal detectable interactions under the conditions tested. Here are the results:

As noted in our response to Reviewer #2, we are currently generating a tagged Firewood knock-in line to enable in vivo co-immunoprecipitation experiments with Fire, Fire-like, and Mvl. These studies will be pursued in a future publication.

Minor comments:

1 Mco4 displayed the highest expression in proventriculus. It was even reported as one of the marker genes for a foregut cell type. Does Mco4 help to reduce iron in proventriculus, or does it play an iron-independent role?

While Mco4 is indeed highly expressed in the proventriculus and has been reported as a marker for a foregut cell type, its functional role in this tissue remains unclear. Whether it contributes to local iron reduction or serves an iron-independent function is an open question that will require further investigation.

2 In Figure 3, panel A shows Prussian blue staining results under two conditions but panel B shows

only one set of data. Please indicate which condition is summarized in panel 3B.

Thank you for pointing this out. We have updated the label in Figure 3B to indicate the condition shown (“+ iron”) for clarity.

3 Do fire and fire-like play slightly different roles? In Figure 4C, fire RNAi induced a phenotype but RNAi against fire-like did not. Do their single knockout lines have reduced staining of BPS-Fe²⁺ precipitates or other phenotypes?

Great question! Due to the compensatory effects between *fire* and *fire-like* (Figure 4H), we generated a double knockout line and focused on their combined function rather than creating single knockouts initially.

However, based on our current data, we anticipate that a *fire* single knockout would exhibit a stronger phenotype than a *fire-like* single knockout. The presence of a phenotype in *tub>fire*-RNAi animals, but not in *tub>fire-like*-RNAi animals (Figure 4C), further supports this expectation.

We are currently planning to generate single knockout lines for *fire* and *fire-like* and will assess BPS-Fe²⁺ precipitate formation and other phenotypes in these backgrounds

4 There are some typos in the text.

e.g. line 363: “BPST”

line 1073: “ph”.

Fixed! Thank you!

Responses to the Reviewers' comments

We thank the reviewers and the editor for their careful reading of the manuscript and for providing constructive feedback. Our responses to the reviewers' comments are shown in blue in this document.

Reviewer #1 (Remarks to the Author):

The authors have addressed all of my comments in their response and the revised manuscript.

Thank you!

Reviewer #2 (Remarks to the Author):

The authors have done an excellent job responding to my comments and have integrated new data when appropriate.

Thank you!

Reviewer #3 (Remarks to the Author):

The authors have addressed our concerns and comments properly. One very minor point that has not been fixed is the typo "BPST" in line 394.

Fixed. Thank you!